# Engaging and disengaging recurrent inhibition coincides with sensing and unsensing of a sensory stimulus

Debajit Saha[1,*], Wensheng Sun[1,*], Chao Li[1,*], Srinath Nizampatnam[1], William Padovano[1], Zhengdao Chen[1], Alex Chen[1], Ege Altan[1], Ray Lo[1], Dennis L. Barbour[1] & Baranidharan Raman[1]

Even simple sensory stimuli evoke neural responses that are dynamic and complex. Are the temporally patterned neural activities important for controlling the behavioral output? Here, we investigated this issue. Our results reveal that in the insect antennal lobe, due to circuit interactions, distinct neural ensembles are activated during and immediately following the termination of every odorant. Such non-overlapping response patterns are not observed even when the stimulus intensity or identities were changed. In addition, we find that ON and OFF ensemble neural activities differ in their ability to recruit recurrent inhibition, entrain field-potential oscillations and more importantly in their relevance to behaviour (initiate versus reset conditioned responses). Notably, we find that a strikingly similar strategy is also used for encoding sound onsets and offsets in the marmoset auditory cortex. In sum, our results suggest a general approach where recurrent inhibition is associated with stimulus 'recognition' and 'derecognition'.

---

[1] Department of Biomedical Engineering, Washington University in St. Louis, St. Louis, Missouri 63130, USA. * These authors contributed equally to this work. Correspondence and requests for materials should be addressed to B.R. (email: barani@wustl.edu).

Sensory systems can rapidly signal the presence of a visual[1,2], auditory[3–5] or an olfactory[6–8] cue encountered by an animal. In addition to being rapid, the stimulus-evoked neural responses are usually elaborate, temporally patterned and tend to outlast the duration of the triggering stimulus[9]. The need for such dynamical neural responses is puzzling, especially considering that the behavioral response initiations can be equally fast, and delayed only by few hundreds of milliseconds after stimulus onset[6]. Further, another bout of strong spiking activities usually occurs after the termination of the stimulus and the behavioral relevance of this 'OFF response' also is not understood. This apparent mismatch between the complexity in the neural encoding and the behavioral decoding raises the following fundamental question: how do neural response dynamics regulate the behavioral responses over time? More importantly, are there general rules of signal processing that are conserved across sensory systems?

A comparison of electrophysiological results reported across sensory systems of different modalities, reveal that there are striking similarities between stimulus-evoked temporally patterned neural responses[9–13]. For example, in the olfactory system, sensory input from olfactory receptor neurons (ORNs) drive spatiotemporal patterns of neural activity in the downstream neural circuits (invertebrate antennal lobe or vertebrate olfactory bulb) that are quite dynamic and information rich at the stimulus onsets and offsets[9,14–17]. In between these transient response epochs, when chemical cues are sustained, the ensemble neural activities in the peripheral and central regions tend to settle down to stable spiking activity patterns, and are often referred to as steady-state responses[9].

Likewise, auditory stimuli elicit onset, steady-state and offset responses from the earliest brainstem nuclei through the auditory cortex. Distinct response patterns exist for specific cell types in subcortical nuclei, where the neuronal dynamics are the fastest[18,19]. Following several synapses of processing, auditory cortical responses also exhibit both transient and steady-state behaviour with a time scale slower than lower processing stations, but closer to the natural dynamics of common sound sources[20–23].

Could these similarities observed in different sensory systems indicate a general framework for encoding/decoding information over time? Here we investigated this issue using two different sensory systems: an invertebrate olfactory system and a primate auditory system. We show that the same sensory circuit can use nearly non-overlapping sets of neurons, and different encoding formats (oscillatory versus non-oscillatory) to represent equivalent information about the identity and intensity of sensory stimulus during different response epochs (at onsets and offsets). Further, our results reveal that switching between distinct neural ensembles over time is temporally correlated with the behavioral dynamics evoked by a stimulus. Notably, our results suggest that such representations provide a potential mechanism for sensory neural networks to meet the evolving demands on the behavioral output during these epochs.

## Results

**Odour-evoked ON versus OFF responses**. We began by examining stimulus-evoked responses of projection neurons (PNs) in the locust antennal lobe (AL) circuit that receive direct sensory input from the olfactory receptor neurons. We used lengthy pulses of odorants (4 s in duration) in order to decouple, and examine the neural responses elicited following the stimulus onset and offset. The stimulus-evoked PN responses could be categorized into two major classes[24,25] (Fig. 1a; also refer Supplementary Fig. 1a–c): increase in spiking activity limited to the periods of odour presentation (ON response), or excitatory responses that occur only in epochs following stimulus termination (OFF response). Consistent with previous findings[14,26], we note that within each PN response category the temporal spiking patterns were heterogeneous.

ON and OFF responses have also been reported in both vertebrate and invertebrate visual systems[27–29]. However, a major difference between visual and olfactory ON and OFF responses is worth pointing out. In the visual system, whether a neuron responds with a light ON or light OFF type response is fixed and the 'cell tuning' does not change in a stimulus-dependent manner[28–30]. On the other hand, in the antennal lobe circuit, we found that an individual PN can respond with either an ON or an OFF response depending on the odour identity and intensity (Fig. 1a). In addition, a comparison of neural firing rates at different processing levels reveals that these OFF responses are weak to non-existent at the level of sensory neurons, but become significant and comparable to the ON responses at the projection neuron level (Supplementary Fig. 2). Therefore, we conclude that the PN response types are not cell-specific, but arise as a result of stimulus-specific circuit interactions within the antennal lobe.

We examined the relationship between the sets of PNs that were activated during stimulus ON and OFF periods (Fig. 1b–e). We found that in general, PNs that were activated during stimulus exposure period were inhibited following stimulus termination with the firing activity reaching below baseline levels (Supplementary Fig. 1a,c). Similarly, the PNs that were activated following stimulus termination were inhibited during stimulus ON period (Supplementary Fig. 1b,c). Therefore, at an ensemble level distinct sets of PNs were activated during odour ON and OFF periods (Fig. 1b–e). In addition, we found that the OFF responses were more distributed over time rather than ON responses that had shorter response latencies (Supplementary Fig. 1d). However, it is worth noting that both in terms of the total number of spikes (across all PNs), and distribution of information rate across neurons (Supplementary Figs 1e,f and 3), both ON and OFF responses were statistically indistinguishable (two-sample Kolmogorov–Smirnov test, $k = 0.1625$, $P < 0.05$, $n = 80$).

Next, we visualized odour-evoked neural activities at an ensemble level by pooling neurons across experiments[9,14,15,17]. Responses were aligned and binned with respect to the odour onset. Subsequently, high-dimensional response vectors were constructed, where each vector element corresponded to the spike count of a single PN in a given time bin (see Methods). To visualize the ensemble neural activity, we performed dimensionality reduction with principal component analysis. We found that each odorant generated two distinct trajectories in the neural response space (Fig. 1b–e): one during stimulus presentation ('ON response' trajectory), and the other following stimulus termination ('OFF response' trajectory). Plots revealing how these trajectories evolve over time are highlighted in Supplementary Fig. 4a. For all odorants examined, we found that the ON and the OFF response trajectories spanned subspaces that were nearly orthogonal to each other (that is, ∼90°). These qualitative results were independently confirmed by computing angular distance between high-dimensional response vectors (Fig. 1f). The generality of these results is shown using a larger odorant panel in Supplementary Fig. 5a,b.

**Comparative analyses of stimulus-evoked ensemble responses**. Our results clearly indicate that responses following the stimulus onset and termination are quite distinct from one another. Therefore, we next sought to examine whether OFF responses have the same specificity as the ON responses. Consistent with

previous results[9,14], we found that both ON and OFF response trajectories changed directions depending on odour identity (Fig. 2a). In comparison, changes in odorant intensity altered the directions of the ON and OFF trajectories only subtly[14], but predominantly lengthened or shrank them (Fig. 2b; Supplementary Fig. 5c). While both the trajectories' span and length increased as odour concentration increased for some odorants (hex, 2oct and iaa), the opposite was true for bzald. These results are consistent with previous findings that both ON and OFF ensemble responses vary with, and therefore contain information about both stimulus identity and intensity[9,14].

How different are the PN combinations activated during the ON and OFF epochs of the same stimulus when compared to PN ensembles activated by different odors or the same odour, but presented at different intensities? Since these neural circuits have been hypothesized to play a pivotal role in discriminating odorants[31–33], we expected different stimuli to activate more distinct combinations of neurons. To understand this and quantitatively compare the similarity between ensemble responses generated in different time bins, we performed a correlation analysis[14] (Fig. 3). As can be noted, the diagonal high-correlation blocks indicate that the ensemble neural activities

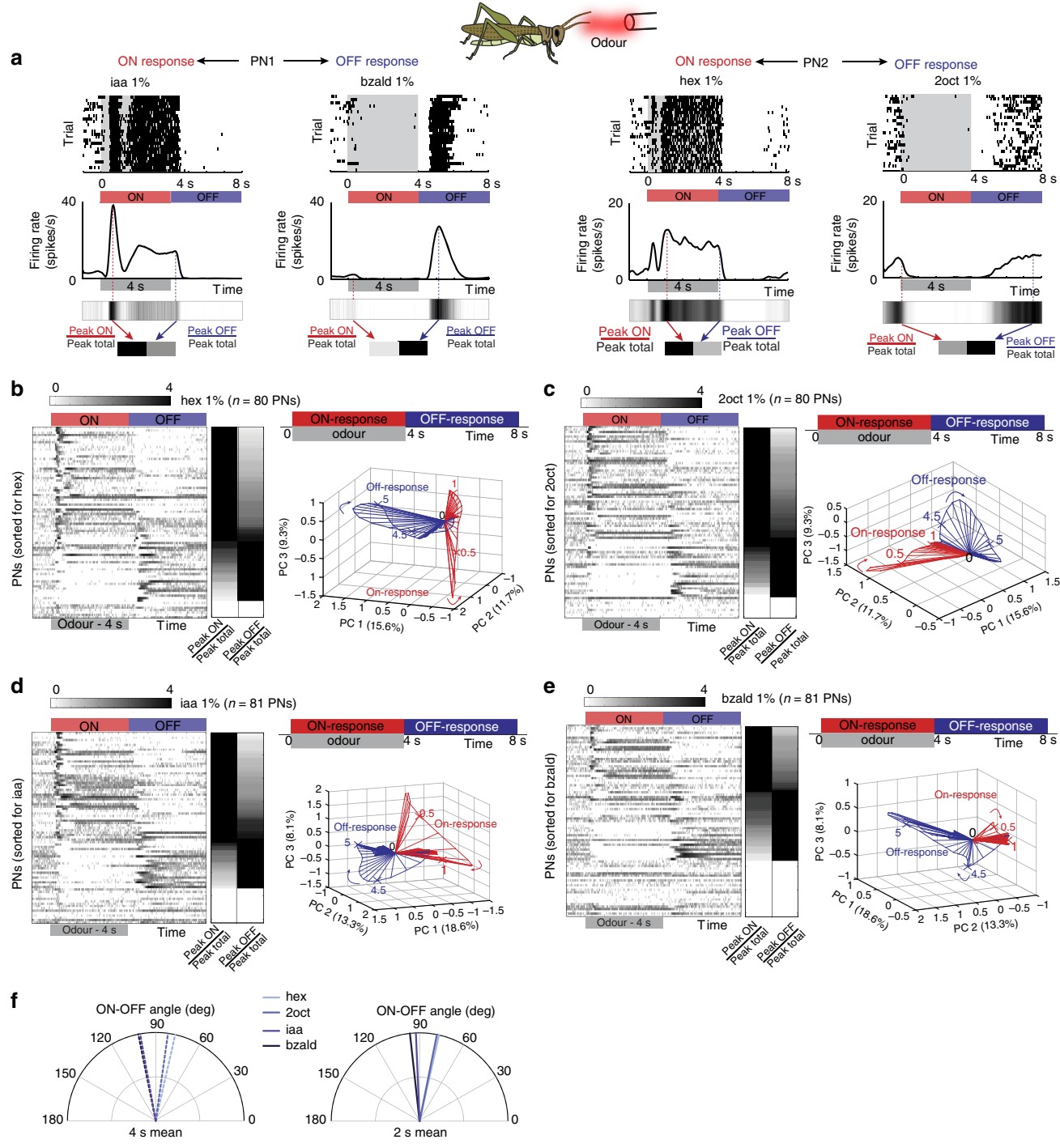

evoked during the stimulus ON periods remained highly similar throughout the stimulus ON period. These high-correlation blocks persisted, albeit to varying levels, even when the comparisons were made between ensemble ON responses evoked by the same odorant but presented at different intensities (Fig. 3c), or between different odorants (Fig. 3d). Similarly, the ensemble neural activities evoked during stimulus OFF periods were highly correlated only among themselves (that is, the lower half of the high-correlation diagonal blocks). The off-diagonal blocks, comparing the ON and the OFF responses were the least correlated in all plots (that is, comparisons between ON and OFF responses of same odorant, different intensities and between different odorants). Furthermore, our results indicate that the combinatorial variations due to stimulus intensity or identity were less drastic, when compared to the differences in the ensemble activities at the

onset and termination of the same stimulus (that is, ON versus OFF responses; Fig. 3e,f; Supplementary Fig. 5d). These results, therefore, reveal that the antennal lobe circuit emphasizes difference between stimulus onsets and offsets better than the dissimilarities between odorants.

**ON versus OFF responses in odour mixtures.** Similar to monomolecular chemicals examined so far, we found that a binary mixture of two odorants also produced ON and OFF responses that were orthogonal to each other (Fig. 4). Predictably, the mixture trajectories appeared to be some combination of the individual odorant responses, both during the stimulus presentation, as well as after the mixture termination. Therefore, the mixture ON trajectories occupied the region between the component ON responses, and the mixture OFF trajectories

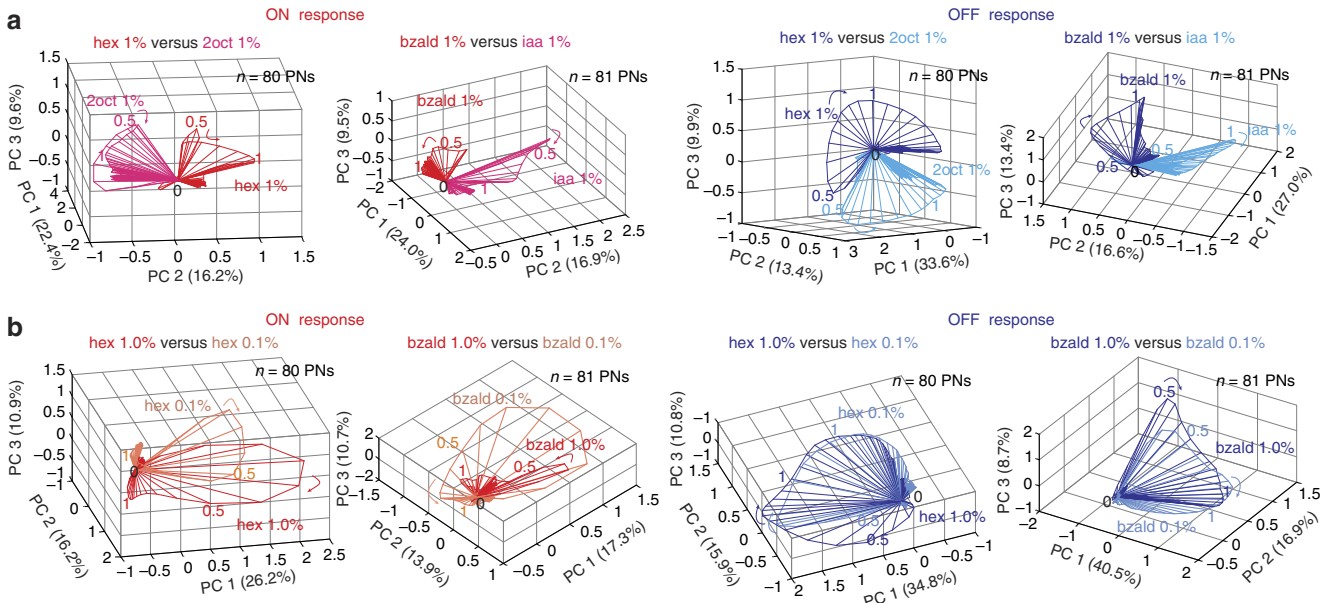

**Figure 2 | OFF responses vary with identity and intensity.** (**a**) Comparison between ON response trajectories evoked by two different odorants is shown after dimensionality reduction for two different odour pairs (left panel; hex versus 2oct and bzald versus iaa). Similar comparison between the OFF response trajectories for the same pairs of odorants are shown in the right panel. The ON and the OFF response trajectories were generated and shown separately for clarity. Note that each odorant evoked a distinct response trajectory during both these epochs. (**b**) Similar plots as shown in **a** but now comparing responses evoked by the same odorant at two different intensities (hex 1% versus hex 0.1% and bzald 1% versus bzald 0.1%).

**Figure 1 | Odour-evoked ON versus OFF neural responses are flexible orthogonal set.** (**a**) Raster plots of spiking activity of two different olfactory projection neurons (PNs; in the insect antennal lobe). Each row corresponds to a single trial, shaded grey box corresponds to 4 s of odour exposure. Twenty-five consecutive trials are shown for each PN. Firing rates in non-overlapping 50 ms time bins (with five point smoothing) are shown below the raster plot. Note there are two prominent PN response categories. (i) ON responses: increase in spiking activity limited to the period of stimulus exposure (PN1–iaa 1% and PN2–hex 1%), (ii) OFF responses: spiking activity is suppressed during odour presentation period but raises above pre-stimulus levels after odour termination (PN1–bzald 1% and PN2–2oct 1%). Same PN can have either ON or OFF response depending on odour identity. Colourbar represents normalized firing rate. The peak responses during stimulus exposure (during ON response window) and following its termination (during OFF response window) are identified. 'Peak total' indicates the maximum response taking into account both epochs. (**b**) Left, Mean PN firing rates (50 ms time bins; averaged across 25 trials) are shown for hexanol (hex) delivered at 1% dilution (v/v). Each row represents the mean firing rate of one PN. Red and blue bars denote Stimulus ON and Stimulus OFF periods respectively. All PN responses are plotted on log scale for comparison. PNs are ordered based on the difference between the peak firing activities observed during the ON and OFF response epochs. Non-responsive neurons or with statistically insignificant responses are shown at the bottom. The normalized peak firing rates during the ON and OFF response periods are shown to the right of the panel. Firing rate increases from light to dark. Non-responsive neurons are shown in white. Right, Olfactory PN spiking activities pooled across locusts are visualized after dimensionality reduction using linear principal component analysis (PCA; see Methods). The percentage of variance captured along the first three principal components is plotted for all PN responses on left. The trajectory traced by the ensemble during the 4 s of stimulus exposure ('ON response') is plotted in red and post-stimulus termination in blue ('OFF response'). Numbers near response trajectories indicate time in seconds since odor onset, and the arrows indicate the direction of evolution over time. (**c–e**) Similar plots as in **b** for three other odorants: 2-octanol (2oct), isoamyl acetate (iaa) and benzaldehyde (bzald). All odorants are at their 1% dilutions (v/v in mineral oil). (**f**) Angular distances between the mean ON and OFF responses of olfactory PNs (high-dimensional vectors of PN spike counts) are shown for all four odorants using two different time windows (4 and 2 s).

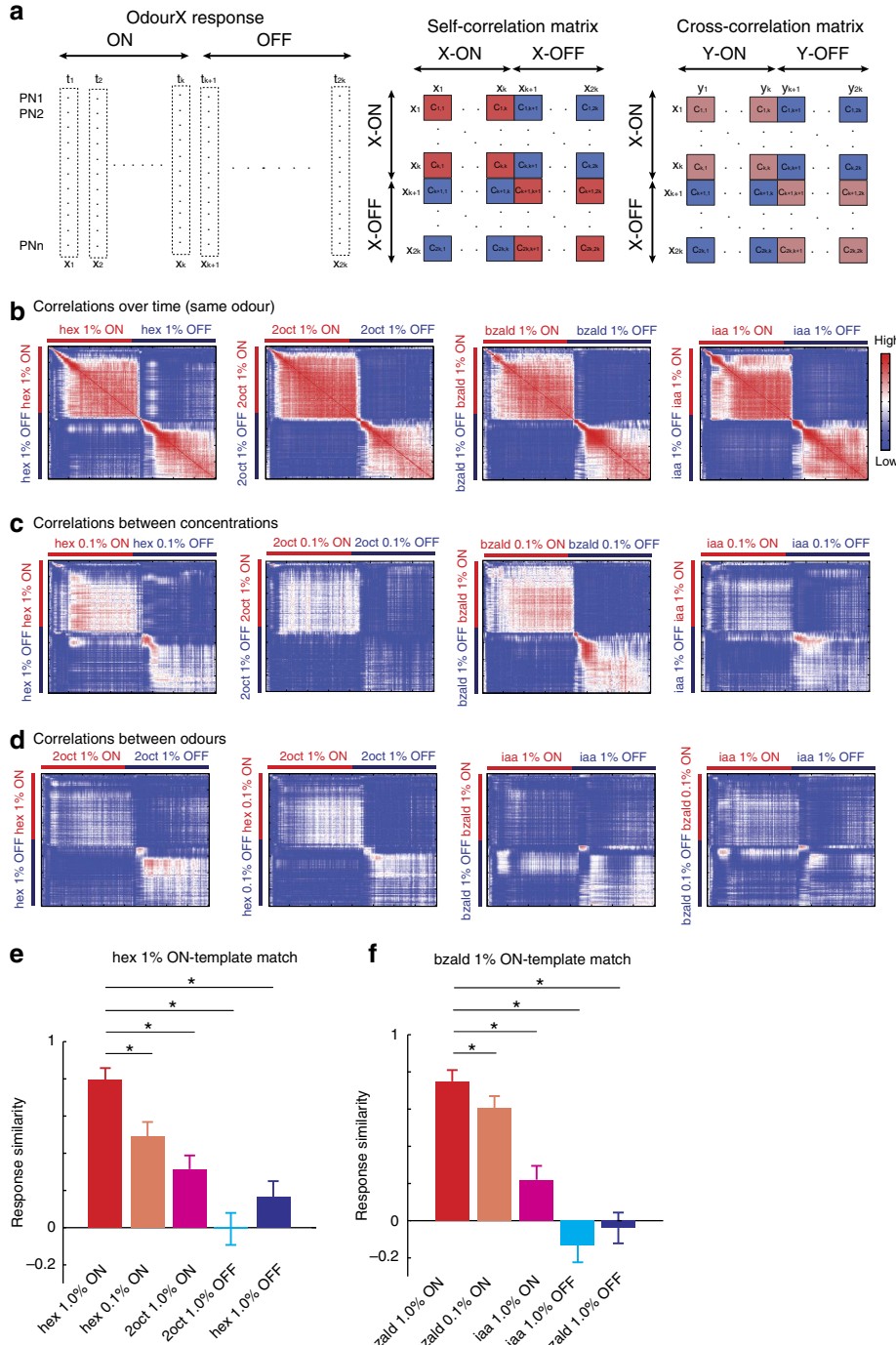

**Figure 3 | ON versus OFF response similarities.** (**a**) Schematic overview of the analysis approach. Each rectangular column indicates population neuron response vector in a 50 ms time bin. Right, self- and cross correlations between response vectors in different time bins were computed and shown as a colour-coded image. (**b**) Correlations between ensemble response vectors evoked by an odorant in different time bins following stimulus onset is shown. The 4 s stimulus ON and 4 s stimulus OFF periods are identified using red and blue bars along the axes. Spike counts were averaged across trials ($n = 25$ trials) and used for this analysis. Note that each pixel represents correlation between one ensemble vector with another. Similarly, one row or column represents the correlation between one ensemble vector with all other vectors in the identified time periods (80 ON response vectors and 80 OFF response vectors). The colour scheme used for representing the correlation values is shown on the right; cooler colours indicate lower correlations; hotter colours represent higher correlations. In general, the diagonal blocks tended to have higher correlations (more red pixels), whereas the off diagonal blocks had pixels mostly of lower correlations (i.e. more blue pixels). (**c**) Similar correlation plots but comparing the ON and OFF response vectors of different concentrations of the same odorants are shown. Comparisons were made between 1% and 0.1% dilutions of the following four odorants: hexanol (hex), 2-octanol (2oct), isoamyl acetate (iaa), and benzaldehyde (bzald). (**d**) Cross-correlations between different odorants are shown. Comparisons were made between the following four odour pairs: hex 1% and 2oct 1%, hex 0.1% and 2oct 1%, bzald 1% and iaa 1%, and bzald 0.1% and iaa 1%. (**e**) A comparison of response similarity (see Methods) between ON and OFF response segments of the same and different odorants are shown. Similarity with respect to hex 1% ON template is shown (mean ± s.d.). Asterisks indicate significant change in similarity (*$P < 0.05$, paired $t$-tests with Bonferroni correction for multiple comparisons, $n = 25$ trials). (**f**) Similar plots are shown but now comparing the response similarity with respect to the bzald 1% ON template.

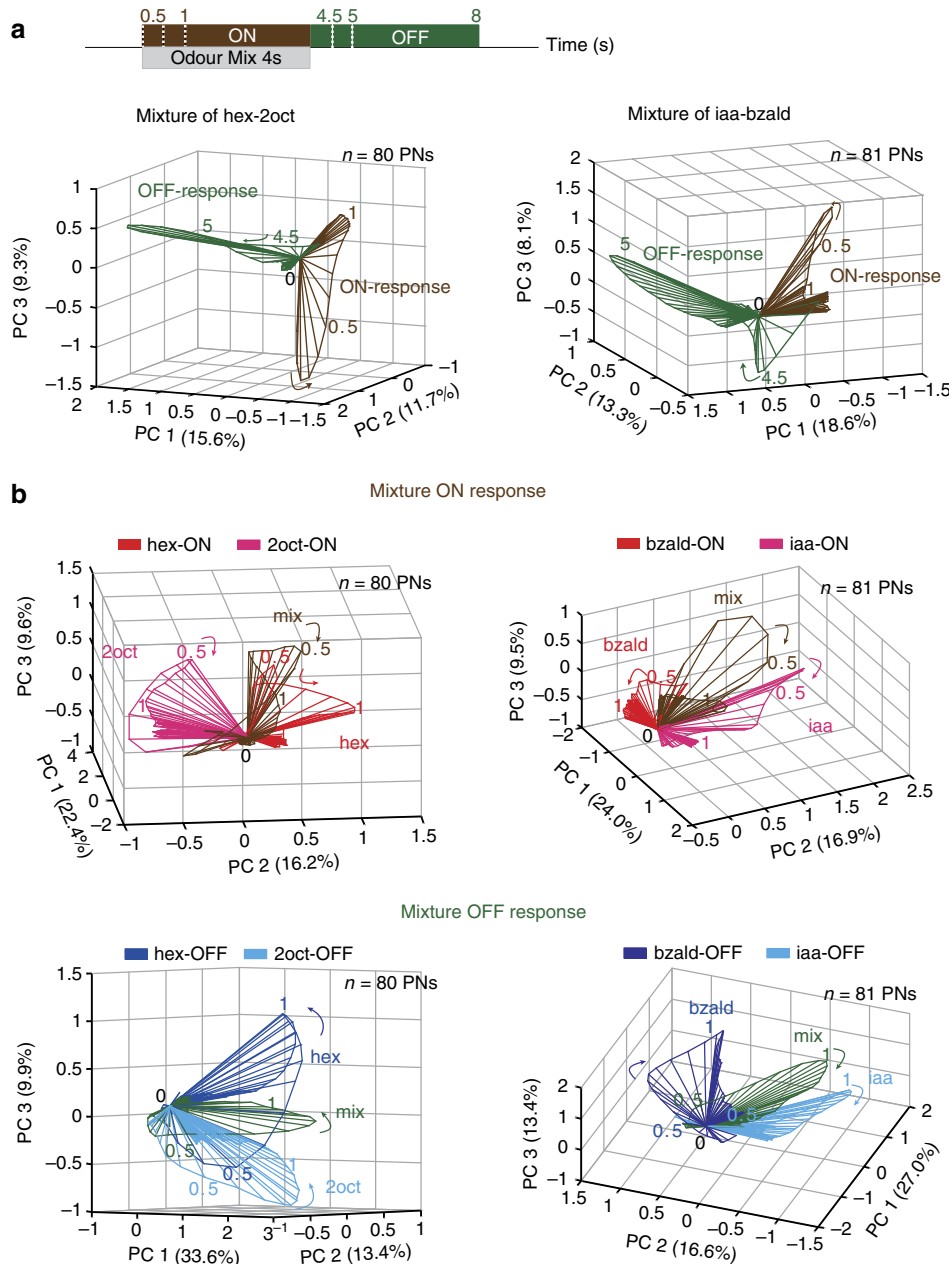

**Figure 4 | Encoding of binary mixtures.** (**a**) Odour trajectories evoked by a binary mixture during ON and OFF epochs are shown after PCA dimensionality reduction. The number of neurons (*n*) used for this analysis and the percentage of variance are shown in each plot. Numbers near response trajectories indicate time in seconds since odour onset. (**b**) Odour trajectories evoked by a binary mixture and its two components are shown after PCA dimensionality reduction. Note, ON responses and OFF responses are compared separately in these plots. Same convention is used as in **a**.

projected onto the space between the OFF responses elicited by each component. These results combined with the results for complex mixtures, such as apple and mint (Supplementary Fig. 5), corroborate our conclusion that these observations regarding ON and OFF responses are general features of odour-evoked neural activities and are not limited to monomolecular chemicals.

**Robustness of OFF responses.** Apart from identity and intensity, naturally encountered odorant plumes also tend to vary in stimulus length[34]. We next examined how invariant were the OFF responses that followed the same stimulus delivered for different durations. We found that the orthogonal relationship between the ON and OFF responses was maintained independent of the

stimulus pulse duration (Fig. 5a). Furthermore, consistent with prior results[9,35], the odour response trajectories for different stimulus durations were well aligned during both the response onsets and offsets. Therefore, the ON and OFF response templates obtained for one odour pulse duration (4 s; see Methods), pattern-matched with ensemble neural activities evoked by the same odorant presented for different durations (Fig. 5b,c). Note that the ensemble response vectors during the entire odour presentation period pattern-matched only with the ON responses, but the response switched and gained similarity with the OFF response template after stimulus termination. On the other hand, different odorants evoked response patterns that were distinct from each other during both ON and OFF response periods (Fig. 5; hex versus bzald).

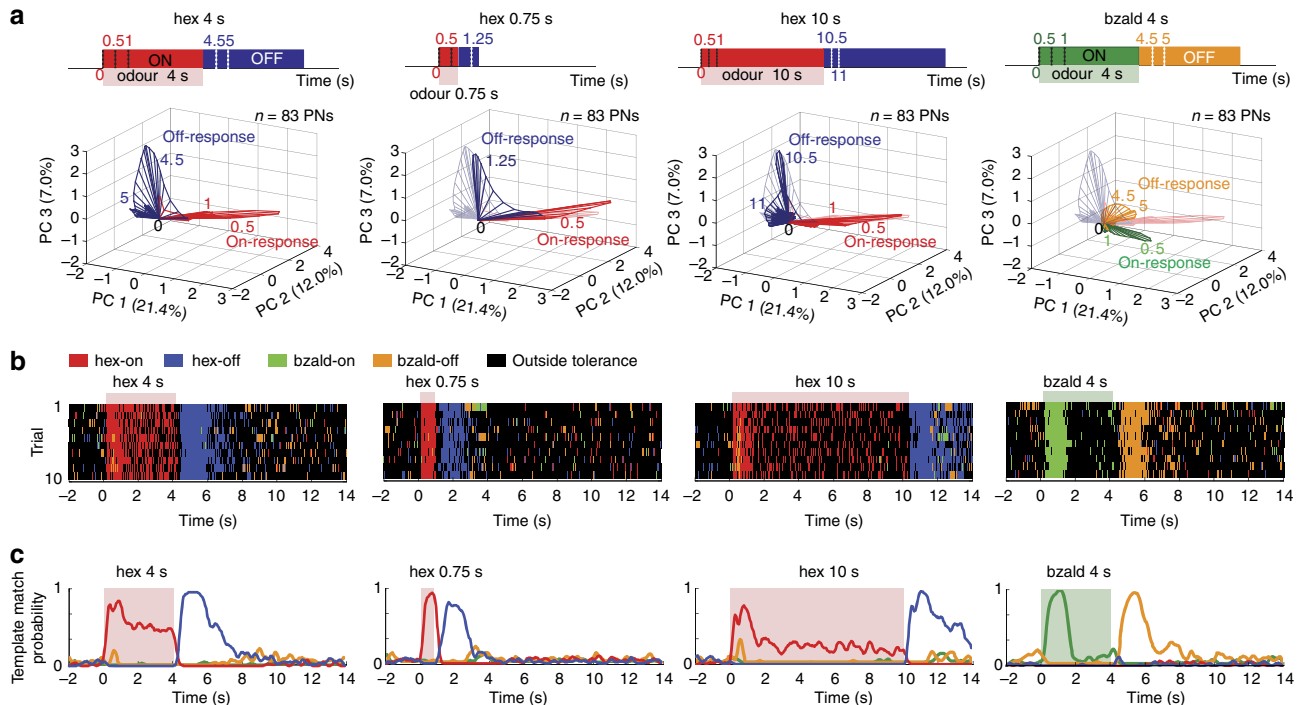

**Figure 5 | Classification analyses of ensemble neural activity.** (**a**) ON and OFF response trajectories evoked by a 4 s hex 1% puff is compared with the response trajectories elicited by a brief (0.75 s) or a lengthy (10 s) presentation of the same odorant, and against a different odorant (bzald 1% delivered for 4 s). For reference, ON and OFF hex (4 s) trajectories are re-plotted in all panels using a lighter shade of red and blue, respectively. *n* represents the total PN number. All other notations and information are consistent with neural trajectory plots shown in Fig. 1b–e. (**b**) Results from a supervised classification analysis (see Methods) are shown. Each row represents a trial and each tick mark corresponds to the class label assigned to the high dimensional neural activity observed in a 50 ms time bin. Each time bin was labelled based on the closest template with which it pattern-matched: hex ON response template—'red tick mark', hex OFF response template—'blue tick mark', bzald ON response template—'green tick mark', bzald OFF response template—'orange tick mark', and time bins when the ensemble neural activity differed significantly from all hex and bzald response templates (> 63°) were labelled using a 'black tick mark'. Classification for ten trials are shown for hex presentations of different durations and a 4 s bzald pulse. A leave-one-trial-out validation was followed for classification of hex 4 s and bzald 4 s conditions. The coloured bar on the top indicates stimulus duration. (**c**) The probabilities of pattern-match with different templates (used in **b**) are shown as a function of time. The coloured box identifies the time period when the stimulus was presented.

In sum, these results suggest that OFF responses are as consistent as the ON responses, and they actively convey information about the termination of a particular stimulus at a specific intensity. Further, our results also show that neuronal networks can use two minimally overlapping sets of neurons to represent equivalent information about a stimulus during different epochs.

**Engaging and disengaging recurrent inhibition.** Apart from using different neural ensembles, are there other differences that distinguish the ON and the OFF responses? To understand this, and to gain mechanistic insights, we made intracellular recordings from GABAergic local neurons (LNs) and cholinergic PNs in the antennal lobe, while simultaneously monitoring the local field potential activity in the mushroom body (the neural circuit downstream to the antennal lobe). Although ON and OFF responsive local neurons have been reported in other model systems[36], consistent with published results in locusts[37], we found that most local neuron responses were limited to the odour onset period. We also found that odour exposures entrained oscillatory activity both in individual local neurons and in the local field potential[38] (Fig. 6a,b). Further, these local neuron responses and field potential oscillatory responses were limited to the odour presentation period (that is, only during ON response epochs; Fig. 6b,c; Supplementary Fig. 6a). Notably, we found that the local

neuron activity remained phase locked with field potential activity only when the stimulus was presented (Fig. 6d). Intracellular recordings from individual PNs were largely consistent with what we had observed in our extracellular datasets and most PN spikes occurred either during or after the stimulus duration (Fig. 6e–g).

We pondered if the recruitment of inhibition could simply arise due to differences in the strength of the odour-evoked responses observed during these epochs. Therefore, we first compared the average spike counts across all recorded PNs during the ON and OFF epochs (Fig. 6h). We found that the PN responses had two distinct peaks, one following odour onset and the other following odour offset. PN spiking activity weakened considerably between these two transient response periods (that is, the sustained/steady-state responses). Interestingly, this weak sustained response was still sufficient to evoke local neuron activity and local field potential oscillations. In comparison, even though the OFF responses were considerably stronger than the neural activity just before the end of the odour pulse, it failed to entrain coherent field potential oscillatory activity (Fig. 6g). Furthermore, as mentioned before, a comparison of cumulative spike counts during the ON and OFF epochs revealed that the spike counts during these time periods were comparable (Supplementary Fig. 1e,f). Therefore, the overall strength of spiking activities across PNs alone appears to be a poor indicator of whether or not the local field potential oscillations are generated by the AL circuitry.

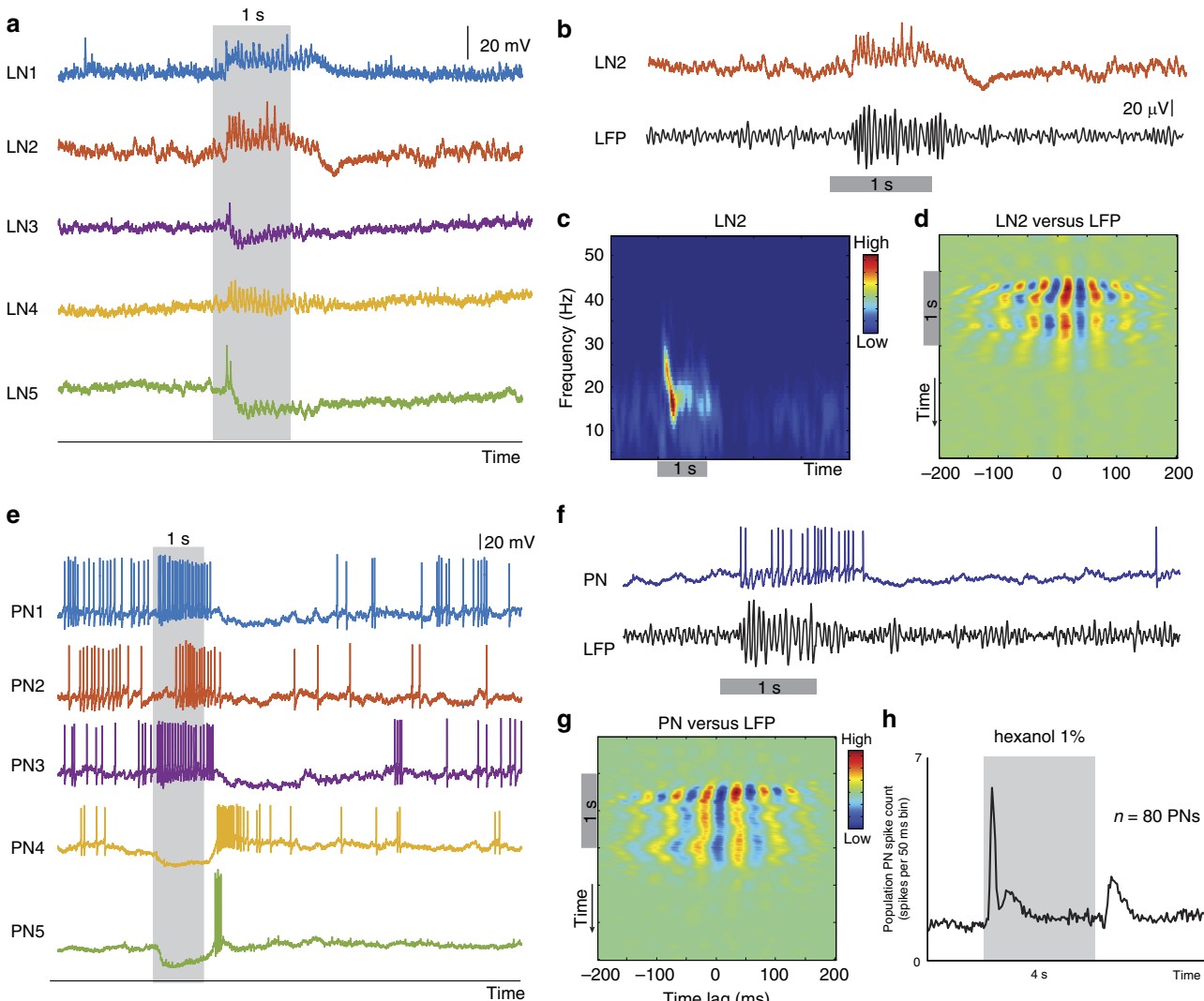

**Figure 6 | Engaging and disengaging recurrent inhibitory network. (a)** Intracellular voltage traces of five different local neurons (LNs) are shown before, during and after a short odour pulse (1 s duration). Note that consistent with previous reports[38], LNs in the locust antennal lobe do not fire full-blown sodium spikes but rather respond to the stimulus with small calcium spikelets. Also, the response to the odour stimulus deviates from baseline levels only during the period of odour exposure and returns back to baseline levels immediately following the odour termination. **(b)** Intracellular response of a local neuron (LN2) and simultaneously recorded extracellular local filed potential (LFP) are shown for a 1 s long odour stimulation. Note that the LN sub-threshold membrane potential fluctuations and LFP oscillation are both limited to the odour presentation window. **(c)** Evolution of power in different frequencies (see Methods) of an odour-evoked LN response (LN2) is shown for three different epochs: before, during and after a 1 s odour puff. **(d)** Cross-correlations calculated between the local neuron membrane fluctuations (LN2) and the local field potential are shown. The alternating peaks (hot colour)/troughs (cool colour) correlation bands can be observed only during the ON response period. Note that the time period of the correlation bands is roughly 50 ms (or 20 Hz). **(e)** Odour-evoked projection neuron (PN) intracellular voltage responses are shown for 5 different PNs. Notice that PNs fired spikes either during the ON or the OFF period in a mutually exclusive manner. **(f)** Intracellular response of a PN and simultaneously recorded LFP signals elicited by a 1 s hexanol pulse are shown. **(g)** Similar plot as in **d** but cross-correlating the PN response with the LFP signal is shown (same data as in **f**). **(h)** Mean spike count across all recorded PNs ($n = 80$) is shown for hex 1% stimulus. Two stimulus-evoked response peaks, one during the ON and the other during the OFF period of odour presentation (4 s duration), can be observed.

In our earlier work[17], we found that the sensory input from ORNs did not have a strong bout of spiking activities after termination of the odorant as was observed in the PNs. Could this difference in the presence/absence of sensory input alone explain the limited entrainment of field potential oscillations during the odour exposure period? First, we note that the presence of strong input from sensory neurons was a good indicator of whether LFP oscillations were present in the AL. However, consistent with the existing data[37], we found that application of picrotoxin, a GABA$_A$ antagonist, alone can reversibly abolish the field potential oscillations (Supplementary

Fig. 6b–e). Note that this pharmacological manipulation did not impact the sensory input to the antennal lobe circuits, but rather blocked the fast inhibition from the local neurons onto the projections neurons. Combining these two observations, we conclude that input from sensory neurons is necessary for recruitment of inhibition from local neurons that then allow generation of oscillatory field potential activity.

In sum, these results indicate that the ON and the OFF responses significantly differ in their ability to engage the local inhibitory circuits, which are necessary for oscillatory synchronization of PN responses. Hence, we conclude that

although the ON and the OFF responses have qualitatively similar information content, their neural encoding formats vary significantly.

**Generating ON and OFF responses in a computational model.** To further understand the mechanisms, we developed a well-constrained computational model of the early olfactory circuits (see Methods, Fig. 7; Supplementary Fig. 7). The AL model had

the following components: (i) feed-forward input from ORNs onto PNs and LNs (ii) recurrent connections between LNs and PNs (iii) a bi-directional adaptive mechanism in individual PNs. Consistent with published results[26,39], we found that recurrent inhibition from local neurons was the essential and sufficient component to generate results similar to our *in vivo* observations: 20 Hz field potential oscillations and phase locking of excitatory and inhibitory ensemble activities only during stimulus ON epoch (Supplementary Fig. 7e,f). Without these recurrent inhibitory

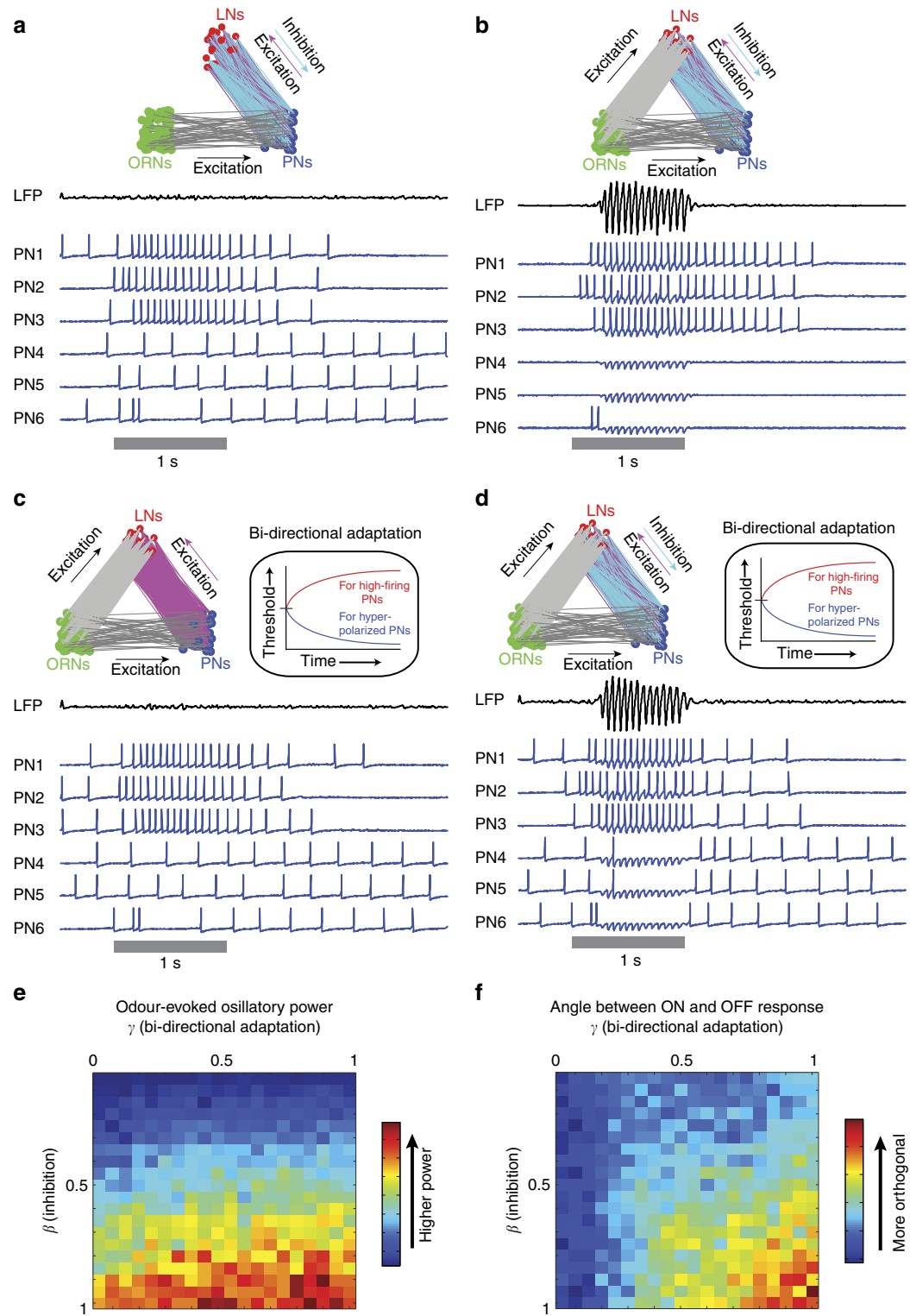

connections from LNs onto PNs, the model did not generate any oscillatory field potential activity (Fig. 7c). This dependence of local neuron activity on stimulus-evoked input limited the recruitment of recurrent inhibition and therefore entrainment of the field potential oscillations to the duration of the stimulus exposure.

While the model with recurrent inhibition alone was sufficient to generate LFP oscillations, as can be noted, the spiking activities in PNs were limited to the epochs, when strong ORN input was available (that is, no OFF responses). Therefore, a bidirectional adaptation mechanism was added to individual PNs that reduced the excitability following high-firing epochs, and at the same time increased the excitability following periods of hyperpolarization. Such adaptive control of neural excitability was necessary to generate orthogonal ensemble activities during stimulus onset and offset (Fig. 7f). This allowed the model to generate the PN OFF responses even when the sensory inputs were decaying back to baseline (that is, weak). We note that to keep the simulations simple and consistent with our electrophysiology data (Supplementary Fig. 2), we did not include any OFF-responsive ORNs in the simulations shown here. In addition, note that the LN inhibition was absent during these periods, as a strong ORN input was necessary in the model to recruit recurrent inhibition. Therefore, the PN OFF responses in the model were also desynchronized, thereby reducing power in the oscillatory field potential activity (Fig. 7d). Bifurcation analyses (Fig. 7e,f) indicated that the strength of local neuron inhibition was the only parameter that controlled the power of the entrained field potential oscillations in the model, whereas a strong inhibition from local neurons and the bi-directional adaptation of PN excitability were both necessary for generating distinct ON and OFF responses.

Hence, our modelling study suggests that stimulus-dependent engagement and disengagement of recurrent inhibition in the antennal lobe circuits provides a simple mechanism for generating distinct ON and OFF neural activities with differing response formats (oscillatory versus non-oscillatory).

**Behavioral relevance of ON and OFF responses**. Are the response patterns observed at the odour onset and offset relevant to odour-evoked behaviour? Earlier studies in rodents and insects have shown that odour recognition can be rapid and usually happens within a few hundred milliseconds of stimulus onset[7,17,40]. On the basis of these results, the early portions of only the ON responses can be expected to play a role in stimulus recognition. What then is the need for another round of stimulus-specific neural activity after odour termination? We sought to examine this issue using an appetitive-conditioning assay. During the training phase, starved locusts were presented with an odorant (conditioned stimulus) followed by a reward (wheat grass; see Methods). We found that locusts reached their asymptotic performance levels after six training trials[17,41]. Following training, locusts were tested in an unrewarded test phase. Locusts that learned the association between the odorant and the reward opened their maxillary palps following the presentation of the conditioned stimulus in anticipation of the reward. Consistent with previous studies, locusts retained the learned association even when tested multiple times in the unrewarded test phase[17,41]. To quantify the behavioral palp-opening response, we painted the distal end of the locust palps with a non-odorous green paint and tracked their whereabouts with fine spatial and temporal resolution (Fig. 8a; see Methods).

We found that the palp-opening responses were quick to start and the palps were kept open, as long as the conditioned stimulus persisted (Fig. 8a). The behavioral responses generalized independent of the duration of the conditioned stimulus (note only a 10 s hexanol pulse was used to train the locusts; see Methods). More importantly, we found that the periods during which the ensemble neural activities pattern-matched with the ON responses corresponded to epochs when the palps were opened and usually kept open (Figs 5c and 8a). In contrast, time segments when the palps closed correlated with those epochs, when neural activity gained similarity with the OFF responses (Figs 5c and 8a). In sum, these results suggest two possible models for translating population neural activity into palp-opening and palp-closing responses: (i) ON model: gaining or losing pattern-match with ON responses underlies palp-opening and palp-closing responses, respectively; and (ii) ON–OFF model: pattern-match with the ON responses triggers behavioral response onset, whereas pattern-match with OFF responses is necessary for terminating the behavioral responses. We found that both the ON and the ON–OFF model could generate predictions consistent with the observed palp opening and closing responses for hexanol presentations of different durations (Fig. 8b,c). Note that we also explored two other model variants for completeness (Supplementary Fig. 8a,b).

**ON–OFF model is a better predictor of behavioral output**. We sought to test these models for translating neural activity to behavioral output by perturbing the pattern-match with the ON responses. To achieve this, we first presented the trained odorant (hex) and a distractant (bzald), as a binary mixture whose components were delivered synchronously (Supplementary Fig. 9). We found that the PN response to this mixture stimulus was dominated by a single component (hex); however, the pattern match with hex ON response templates (solitary presentations)

**Figure 7 | Modelling of ON–OFF neural activity.** Local field potential activity (LFP; top trace) and six modelled projection neuron (PN) spiking activities are shown. Four different model architectures were evaluated: (**a**) Model architecture 1: feed-forward ORN inputs to the local neurons (LNs) were removed. This made the total input received by LNs too weak and therefore the LNs were not activated when stimulus was introduced. As a result, PNs did not receive any feedback inhibition. Also note that the stimulus-evoked oscillatory field potentials were not observed. (**b**) Model architecture 2: LNs received inputs from both ORNs and PNs. As a result, LNs were activated and PNs received recurrent inhibition from LNs. Oscillatory field potentials were observed in this model during stimulus exposure period. However, the model did not generate a strong activity following stimulus termination (that is, no 'OFF' responses). (**c**) Model architecture 3: PN excitability was adapted in a bi-directional manner. LN inputs to PNs were removed. PNs did not receive feedback inhibition. Therefore, the model did not evoke stimulus-evoked LFP oscillations, or strong PN responses following stimulus termination. (**d**) Model architecture 4: PN responses were adapted in a bi-directional manner. LNs received inputs from ORNs and PNs. Therefore, LNs were activated by input stimulus and PNs received feedback inhibition. Therefore, the model produced stimulus-evoked oscillatory field potentials. The strong inhibition to a subset of PNs during the odour input increased the excitability of the inhibited PNs and thereby causing a strong OFF response in this model. (**e**) Bifurcation analysis showing the relative importance of recurrent inhibition from LNs (*y*-axis) and bi-directional response adaptation (*x*-axis) for generating oscillatory local field potential in the 5–55 Hz frequency range. The horizontal banding reveals that the strength of the feedback inhibition alone is necessary and sufficient for generating LFP oscillations. (**f**) Bifurcation analysis showing the relative importance of recurrent inhibition from LNs (*y*-axis) and bi-directional spiking threshold adaptation (*x*-axis) for generating distinct ON and OFF neural activities. Note that both strong recurrent inhibition and bi-directional spiking threshold adaptation are important for generating a distinct ON versus OFF responses.

was diminished (Supplementary Fig. 9a,b). Matching these classification analyses results, we found that locusts trained with hex, responded to the binary mixture of hex and bzald with a similar reduction in POR (Supplementary Fig. 9c). These results clearly demonstrate that reduction of pattern match with the ON response template of conditioned stimulus diminishes the behavioral POR responses.

Next, we presented the same two odorants in series such that onset of the distractant (bzald; untrained odorant) happened 500 ms before the termination of the conditioned stimulus (hex; Fig. 8d). Consistent with previous findings[17], and unlike the synchronously presented binary mixture case, the neural activity

remapped to gain pattern-match with the second odorant in the sequence (that is, bzald) following its onset. However, following the termination of the conditioned stimulus (hex), we found that the ensemble neural activity again remapped to gain similarity with the OFF response of the first odorant (that is, hex; Fig. 8d; Supplementary Fig. 10a).

Interestingly, we found that the palp-opening response to hexanol (the conditioned stimulus) did not end, when a distractant was introduced (Fig. 8e). Rather the closing of palps began after the termination of the conditioned stimulus following epochs, when pattern-match with hexanol OFF responses was observed (Fig. 8e). Therefore, the amount of time it took for

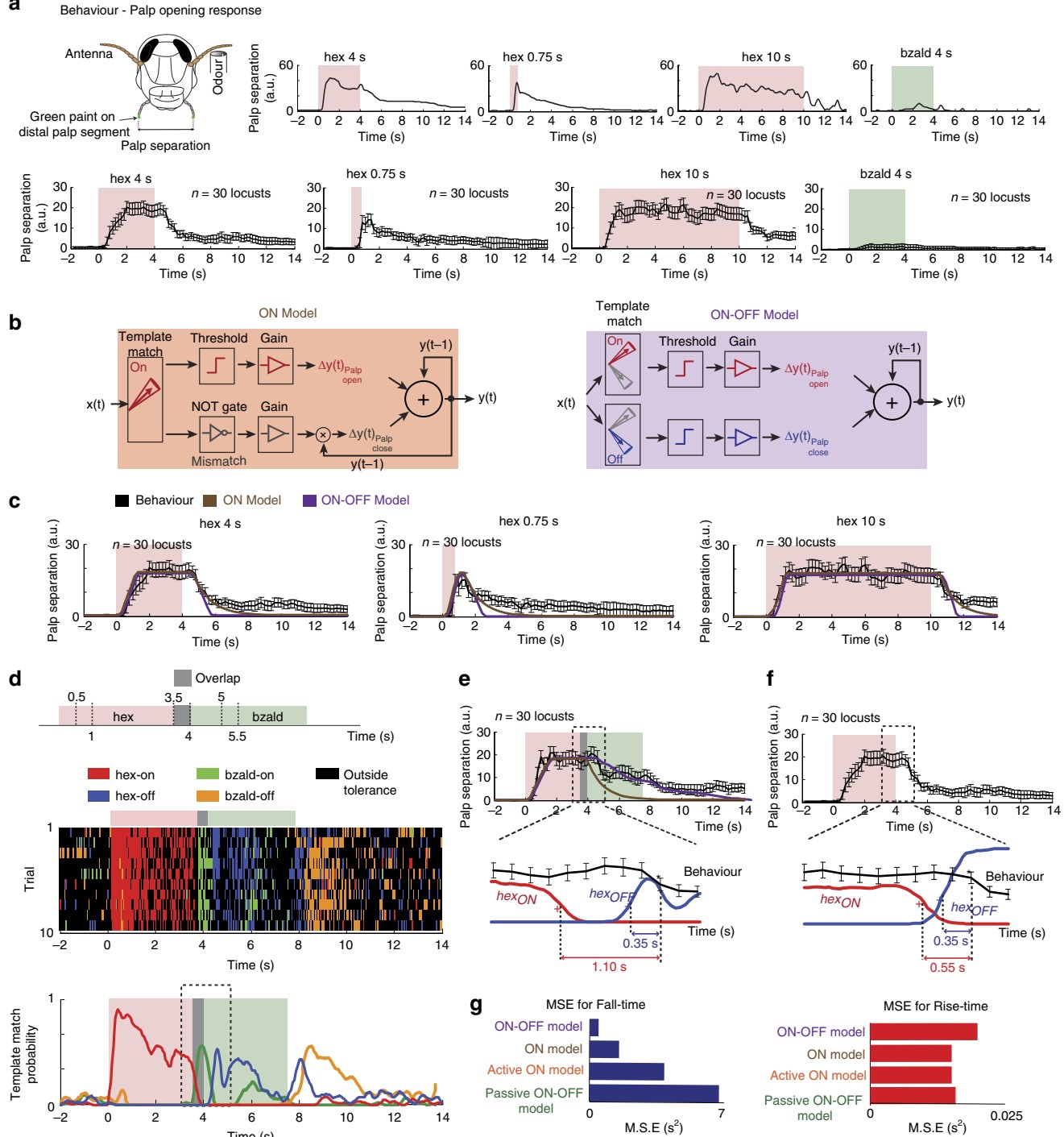

closing palps following a mismatch with the ON responses doubled across the two conditions tested (Fig. 8e,f).

Two other observations are worth pointing out here. First, the distractant presented solitarily did not evoke a significant palp-opening response (Fig. 8a). So the prolongation of the POR after introduction of bzald cannot be explained based on the ongoing PN activity during this epoch, as its pattern-matched with the bzald ON template (Fig. 8d). Second, it is worth noting that in the overlapping sequence, the degree of pattern match with the hex OFF response template was diminished due to the presence of the distractant. Matching this physiological result, we found that the POR response termination was also slower than that observed in the case of solitary hex presentations (Supplementary Fig. 10b,c). Therefore, these results suggest that after the palps have been opened, a pattern-match with the ON responses may not be necessary for sustaining the behavioral response. More importantly, gaining pattern-match with the OFF responses of the conditioned stimulus is a good indicator of the palp-closing response dynamics. This interpretation is supported by the modelling results, which revealed that only the ON–OFF model was able to generate consistent palp-closing behaviour across conditions (Fig. 8g, Supplementary Fig. 8c). Furthermore, we found that POR to the hex-0.5s overlap-bzald stimulus sequence can be better predicted using results from the classification analysis than those directly made using the POR data. In other words, the time series of ensemble neural activities was Granger causal with the behavioral POR evolution over time ($f_{Neural->POR} = 7.37$, $P < 0.05$, $n = 80$; see Methods).

Finally, to further confirm this hypothesis, we presented the conditioned stimulus (hex) in a pulsatile fashion (Fig. 9). We found that the ON and OFF responses precisely encoded the presence and absence of the hexanol puffs. As can be predicted, trained locusts opened or closed palps during epochs when hex ON and OFF responses were observed, respectively. Taken together, these results strongly support our hypothesis that orthogonal neural activities may underlie opposing behavioral responses in this olfactory system.

**ON versus OFF responses in the marmoset auditory cortex.** Finally, we wondered how general are these signal processing features observed in the insect olfactory system? To understand this, we examined the response of cortical neurons in the marmoset monkey auditory cortex (A1 area) to monotones (0.5 s duration) (Fig. 10). Analogous to results in the olfactory system, we found that the same neuron could respond with an ON or an OFF response to monotones depending on the monotone frequency (Fig. 10b). In addition, it might be worth noting that primary auditory neurons in mammalian auditory system do not have the ON and OFF opponent responses observed in the visual system[11,42,43]. Therefore, we conclude that A1 neurons are 'network tuned', as in olfaction and not 'cell tuned' as in vision.

At the ensemble level, we found that cortical neuron responses to prolonged monotones were not sparse and a large fraction of the recorded neurons had a statistically significant response, when both ON and OFF response epochs were considered (Fig. 10c–h, left panels). Minimally overlapping ensembles of neurons were activated by the same sound following its onset and offset. This again resulted in two orthogonal neural response trajectories for each monotone, one following the sound onset to encode its presence and the other following sound offset to indicate its termination (Fig. 10c–h, right panels, Fig. 10i). In sum, these results reveal that locust olfactory circuits and marmoset auditory cortical circuits may employ conserved processing principles to actively encode stimulus presence and termination.

**Discussion**

A behavioral response initiated by any sensory stimulus has to be reset after its termination. In most cases, the response onset (deviation from baseline) following the stimulus introduction, and the reset (return to baseline) following its termination are by necessity opposites of one another. Is the behavioral response reset actively brought about by the neural circuitry, or is it a result of a passive return of stimulus-evoked activity to the spontaneous level? Two lines of evidences appear to suggest that a more direct representation of the stimulus absence will be necessary in most sensory systems. First, sensory memory following stimulus encounters may persist even after the termination of the stimulus[26,44–46]. Second, in natural settings, sensory cues are mostly encountered in overlapping sequences, and a passive return to baseline may not happen until after all of the succeeding stimuli terminate. Furthermore, in sensory systems, absence of a stimulus can be as informative as their presence (light versus dark[47,48] or heat versus cold in temperature sensing[49,50]). Taking into account that most sensory stimuli generate another round of transient activity following stimulus termination[9,11,16], it would appear that an active signal regarding the absence of stimulus is available in many sensory systems.

**Figure 8 | Stimulus-evoked OFF responses are required for behavioral reset.** (**a**) Top left panel, schematic of palp-opening response (POR) behaviour in trained locusts following presentations of the conditioned stimulus. Separation between the maxillary palps was used to quantify behaviour. Top right panels, POR of a single locust to the conditioned stimulus (hexanol, pink) for three different durations and to an untrained stimulus (4 s pulse of bzald, green). Bottom panel, median POR for all locusts (± s.e.m., $n = 30$ Locusts). (**b**) Schematic of ON model (left panel) and ON–OFF model (right panel) to predict palp opening and closing responses from ensemble activity. (**c**) Behavioral responses are predicted from both ON model and ON–OFF model (PORs are re-plotted from **a**). Models were fit only using the 4 s POR data (see Methods). (**d**) Top overlapping sequence of hex and bzald was presented. bzald was introduced 0.5 s before the termination of hex. Middle, classification analysis for the ensemble activities generated by the hex-bzald overlapping sequence. ON and OFF responses observed during solitary hex and bzald were used as templates to be pattern-matched (same templates as used in Fig. 5b). Bottom, pattern-match probabilities with different response templates as a function of time. Boxed region identifies a small time segment starting just before bzald onset and ending after the termination of hex. (**e**) POR (median ± s.e.m., $n = 30$ Locusts) for same overlapping sequence of hex and bzald (in black). ON- and the ON–OFF model fits are shown (same colour code as in **c**). The inset (dotted region) magnifies the epochs before, during and after hex-bzald overlap. The palps closed not when the pattern-match with the ON response of the conditioned stimulus was lost (that is, red curve returns to baseline), but when similarity with the OFF responses was gained (that is, blue curve ramps up from baseline). A Wilcoxon signed-rank test was used to detect the time bin when the first significant reduction in palp-opening response occurred (black star; $P < 0.05$ and Bonferroni corrected for multiple comparisons). The peak derivative of pattern-match probabilities was used to determine when a mismatch with ON responses (red cross) or a match with OFF responses began (blue cross). (**f**) the behavioral response to solitary presentations of hex and the evolution of neural response pattern-match over time are shown. The latency of palp closing following loss of pattern-match with the hexanol ON responses doubled for the hex-bzald overlapping stimulus sequence while time from the pattern-match with the OFF responses to the onset of palp-closing behaviour was constant. (**g**) A quantitative comparison of the four different models. The mismatches of rise and fall time constants predicted by these models with actual behavioral results for different hexanol presentations are quantified as mean-squared errors.

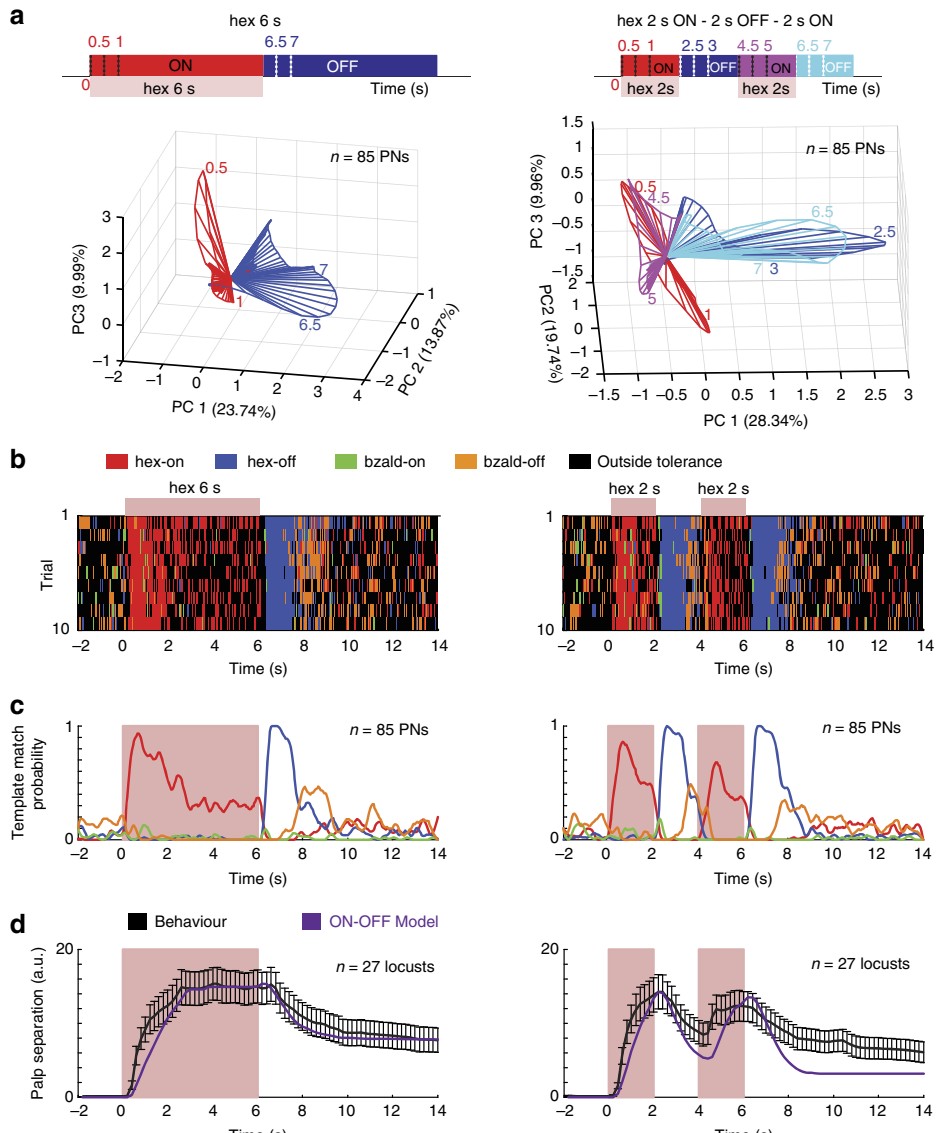

**Figure 9 | Neural and behavioral responses to odour pulses. (a)** Left, ensemble PN response trajectory is shown for 6 s long hexanol pulse. ON and OFF trajectories are identified using red and blue colours, respectively. Right, PN response trajectory is shown for 2 s ON–2 s OFF–2 s ON hexanol pulse. Red and purple portions of the trajectories indicate the ensemble ON activity during the first and second hexanol pulse (ON responses), and the blue/cyan trajectories trace the PN ensemble responses following the termination of the first and the second hexanol pulse. **(b)** Results from a classification analysis are shown in a bin-by-bin, trial-by-trial fashion. Based on the closest template, a class label has been assigned for each 50 ms time bin. hex ON response template—'red tick mark', hex OFF response template—'blue tick mark', and time bins when the ensemble neural activities that differed significantly from both hex ON and OFF response templates were labelled using a 'black tick mark'. Classification for ten trials are shown for 6s hex pulse and 2 s ON–2 s OFF–2 s ON hexanol pulse. A leave-one-trial-out validation was used for generating these results. The coloured bar on the top indicates stimulus exposure periods. **(c)** The probabilities of pattern-match with hex-ON and hex-OFF templates are plotted as a function of time for a uninterrupted 6 s hexanol pulse and a 2 s ON–2 s OFF –2 s ON hexanol pulses. **(d)** Behavioral palp opening responses are plotted (median ± s.e.m., $n = 27$ Locusts) for hex 6 s and hex 2 s ON–2 s OFF–2 s ON pulses. The prediction from the ON–OFF model (purple trace) is also shown for comparison.

For the OFF responses to encode stimulus absence, the neural activities during this epoch must be different from the ON responses and exclusively encode for each stimulus. Our results indicate that the ensemble neural activities at sound and odorant offsets were nearly orthogonal to (that is, independent from) the ON responses. Nevertheless, both these neural activities during stimulus onsets and offsets were able to uniquely encode for identity and intensity of a sensory cue. Importantly, while the onset responses were necessary for the initiation of the behavioral response ('the palp-opening response'), our results reveal that the offset responses are necessary to actively terminate it ('the palp-closing response'). Thus, orthogonal neural activities

encoded for the presence and absence of a stimulus, and were translated to generate behavioral responses that were opposites of one another (start/onset versus stop/reset). Such mapping of distinct neural activities to generate behavioral responses that are opposites of one another have indeed been shown in a number of neural systems[27,49–56]. Our work is the first to show that a single sensory stimulus can activate unique and independent sets of neurons during and after its presentation in order to meet opposing behavioral output demands during these epochs.

Significantly, our results from the invertebrate olfactory circuit and the primate auditory cortical circuit are strikingly similar. Considering that these two sensory circuits differ in their

modality, complexity, evolutionary origin, and their position in the sensory information processing hierarchy, we speculate that our results reveal a conserved information processing approach to signal stimulus presence and absence in a wide variety of sensory systems.

## Methods

**Odour stimulation.** Neat odour solutions (Sigma-Aldrich) were diluted in mineral oil to their 1 or 0.1% concentrations by volume (v/v). The diluted odour solution was placed in a 60 ml sealed glass bottle with separate inlet and outlet lines. A pneumatic pico-pump (WPI Inc., PV-820) was used to displace a measured volume of the odour-bottle headspace ($0.1\,l\,min^{-1}$) that was then injected into a

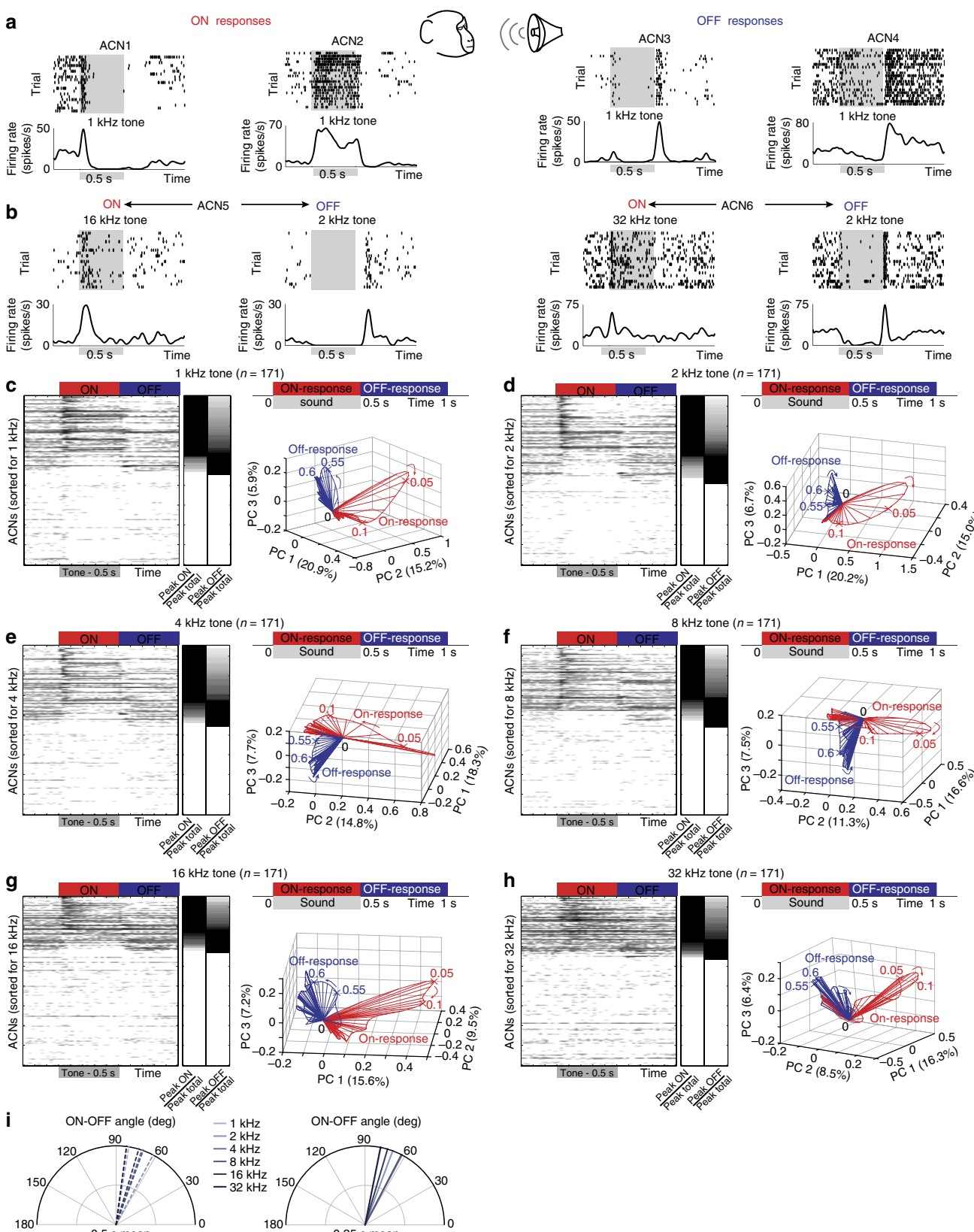

desiccated and filtered carrier air stream ($0.75\,l\,min^{-1}$) directed towards the locust antenna. A vacuum funnel was placed right behind the locust antenna to provide a constant flux and ensure removal of the delivered odour vapours. Each odorant was presented in a pseudorandom manner (blocks of 10 or 25 trials) with 60 s inter-trial intervals and 15 min inter-block intervals. The following odorants were used in this study: hexanol (hex), 2-octanol (2oct), isoamyl acetate (iaa), benzaldehyde (bzald) and binary mixtures of hexanol–2-octanol and isoamyl acetate–benzaldehyde. Each odorant was delivered at their 0.1% or 1% dilutions (v/v). The pre-mixed binary mixture contained vapours of individual components each at 0.5% (v/v) dilution levels.

**Olfactory electrophysiology.** Young locusts (*Schistocerca americana*) with fully developed wings (post fifth instar) of either sex were selected from a crowded colony. Locusts were immobilized with both antennae intact, and then the brain was exposed, desheathed, and continually perfused with locust saline as demonstrated previously[38,57,58]. *In vivo* extracellular recordings from the antennal lobe were performed using 16-channel, $4 \times 4$ silicon probes (NeuroNexus). Electrodes were gold plated such that their impedances were in the 200–300 kΩ range. The extracellular signals were acquired using a LabView data acquisition system. Raw extracellular signals were collected at 15 kHz sampling rate, amplified at 10 k gain using a custom made 16-channel amplifier (Biology Electronics Shop; Caltech, Pasadena, CA), filtered between 0.3 and 6 kHz before spike sorting.

Intracellular recordings were performed as previously described[35]. Briefly, sharp glass micropipettes were filled with 0.5 M potassium acetate solution to achieve impedance in the range of 50–150 MΩ. Voltage signals were amplified (Axoclamp-2B, Molecular Devices) and saved at 5 kHz sampling rate using Labview. Local field potentials were recorded simultaneously using saline-filled glass microelectrodes (4–10 MΩ) and low-pass filtered using a DC amplifier ($<100\,Hz$, Brown-Lee Model 440)[59].

**Olfactory neural datasets.** Recordings for different pairs of odorants (*dataset1*: hex–2oct and *dataset2*: iaa–bzald, $n = 25$ trials each) were made separately (that is, different sets of PNs). In addition, PN recordings to monitor the responses to different durations of hexanol and to characterize the response patterns evoked by overlapping presentations of hexanol and benzaldehyde were collected separately (*dataset3*: hex durations and hex-bazld overlap, $n = 10$ trials). A fourth PN response was collected to compare ensemble responses to six puffs of hex and bzald, a binary mixture of those two chemicals, and a 2s ON—2s OFF—2s ON pulsatile delivery of hexanol (*dataset4*: hex and bzald, $n = 10$ trials). In all the neural datasets, the delivery of odorants was pseudo randomized in blocks of 25 or 10 trials.

To test the generality of our results, we reanalyzed a previously published dataset[17] of PN responses to a wide range of analytes (only used in Supplementary Fig. 2).

**PN spike sorting.** Spike sorting was done using a conservative approach described in earlier works[8,17,60]. In brief, the following criteria were used for the single-unit identification: cluster separation $>5$ noise s.d., number of spikes within $20\,ms < 6.5\%$, and spike waveform variance $<6.5$ noise s.d. Using this approach, a total of 329 PNs were identified from 41 locusts.

- 80 PNs were identified for hex-2oct pair (used in Figs 1–4);
- 81 PNs for bzald-iaa pair (used in Figs 1–4);
- 83 PNs for hex durations and hex-bzald sequence (Figs 5 and 8);

- 85 PNs for hex-bzald mixture and pulsatile stimulation (Fig. 9 and Supplementary Fig. 9).

**Sorting of PN response.** PN responses were sorted in Fig. 1b–e based on the following metric:

$$\text{Response Difference} = \frac{\text{norm Peak ON} - \text{norm Peak OFF}}{\text{norm Peak ON} + \text{norm Peak OFF}}$$

Nonresponsive neurons were identified and moved to the bottom of these plots. Nonresponsive neuron criterion was similar to that used before[17] with only exception that here the entire time window involving both the ON and the OFF responses was taken into consideration.

**Dimensionality reduction analysis for PN responses.** We used the linear principal component analysis technique for the purpose of visualizing high-dimensional neural response trajectories. For this analysis, we binned the PN responses in 50 ms non-overlapping time bins and averaged the responses in a given time bin across trials. This resulted in a time series matrix of $n$ neurons (rows or dimensions) and $m$ time steps (columns) for each odorant. When comparing trajectories elicited by different odorants, data matrices obtained for different odorants were concatenated to increase the number of columns. The resulting high-dimensional vector in each time bin was projected along the eigenvectors of the $n \times n$ response covariance matrix. Low-dimensional data points that represented response vectors in adjacent time bins were connected to generate low-dimensional response trajectories. Finally, the response trajectories were smoothed using a three-point moving average filter.

A time window that included both ON and OFF responses (twice the length of the odour pulse) was used while comparing response trajectories evoked following stimulus onset and termination (Figs 1b–e,4a and 5a, Supplementary Figs 4,5a). For comparison of ON or OFF responses evoked by different odorants and by the same odorant at different concentrations, only a 4 s window comprising of either the ON or the OFF responses were exclusively used (Figs 2a,b and 4b, Supplementary Fig. 5c). This was required as differences between the ON and OFF responses of the same stimulus was the dominant source of variance in the data set.

Note that this dimensionality reduction analysis was used only for qualitative purposes. All quantitative analyses were performed using the high-dimensional PN response vectors.

**Angle between mean ON and OFF projection neuron responses.** High-dimensional PN response vectors were generated using all recorded neurons. The mean baseline response during a 2 s pre-stimulus period, immediately preceding stimulus onset, was subtracted from all response vectors. The high-dimensional response vectors were averaged over the entire duration of the odour pulse (4 s) to generate the mean ON response template ($W_{ON}$). Similarly, the high-dimensional vectors were averaged for a 4 s period following the odour pulse termination to generate the mean OFF response template ($W_{OFF}$). The angle between the mean ON and OFF responses were computed as follows:

$$\text{angular distance} = \cos^{-1}\left(\frac{W_{ON} \cdot W_{OFF}}{|W_{ON}||W_{OFF}|}\right) \qquad (1)$$

Different analysis windows (2 s and 4 s) were used to ensure that the orthogonal relationship between these two response templates were insensitive with respect to the time bin size (Fig. 1f; Supplementary Fig. 5b).

**Figure 10 | Sound-evoked ON versus OFF neural responses in the marmoset auditory cortex.** (**a**) Spiking activities of four different primary auditory cortical neurons (A1, marmoset monkeys) are shown as raster plots. Each row corresponds to a single trial, which includes a 0.3 s pre-stimulus period, 0.5 s stimulus exposure period (shaded gray region; 1 kHz tone), and a 0.7 s post-stimulus period. Firing rates in 50 ms overlapping Gaussian windows are shown below the raster plot. Left, two ON responsive neurons are shown (ACN1 and ACN2). Increase in spiking activity was limited to stimulus periods. Right, two OFF responsive neurons are shown (ACN3 and ACN4). (**b**) Same auditory cortical neuron can respond during either the ON or OFF period of sound presentation depending on the monotone frequency. The raster plots and the firing rate plots are shown for two ACNs following the convention in **a**. (**c**) Left, Individual auditory cortical neurons' mean firing rates (25 ms time bin, averaged over 20 trials) to a 1 kHz monotone stimulus are shown on log scale. The grey bar indicates the stimulus duration (0.5 s). Each row corresponds to mean firing rate of one neuron, red bar indicates ON period and 0.5 s post-stimulus period is indicated by blue bar (OFF period). All recorded ACN responses are shown. Non-responsive neurons are plotted at the bottom of each panel, while responsive ACNs are sorted based on the difference between the normalized peak responses for ON and OFF periods of the sound presentation (see Methods). Normalized peak firing rates for ON and OFF periods of each ACN are shown at the right side of the panel. Normalization was performed against the maximum response observed across both epochs (Peak total). Firing rate increases from light to dark. Non-responsive neurons are shown in white. Right, Auditory cortical neuron spiking activities pooled across experiments are visualized after PCA analysis (see Methods). The percentage of variance captured along each principal component is identified along each axis. $n$ denotes number of neurons in analysis. The trajectory traced by the ensemble neural activities during the 0.5 s of stimulus exposure ('ON response') is plotted in red. To provide contrast, the 0.5 s of neural activities following stimulus termination ('OFF response') is plotted in blue. Numbers near response trajectories indicate time in seconds since sound onset, and the arrows indicate the direction of trajectory evolution over time. (**d–h**) Similar plots as in **c** but showing ensemble neural activities evoked by monotones of five other frequencies (2, 4, 8, 16, 32 kHz respectively). (**i**) Angular distances between the mean ON and OFF responses of auditory cortical neurons (high-dimensional vectors of ACN spike counts) are shown for all six monotones using two different time windows (0.5 and 0.25 s).

For Fig. 3e,f and Supplementary Fig. 5d, the comparisons (cosine of the angle obtained from equation 1) were made either between mean ensemble activities (2 s window) during different epochs of a single stimulus (ON versus OFF), or between the ON and OFF responses evoked by two different stimuli.

**Classification analysis.** A bin-by-bin, trial-by-trial classification analysis[8,17] was used to determine the pattern-match between PN responses observed in a particular time bin with the ON and OFF response templates of a particular odorant (Figs 5b,8d and 9b, Supplementary Fig. 9b). Note that the ON and OFF templates were generated using solitary hexanol or benzaldehyde exposures using spike counts in the 2 s time windows immediately following stimulus onset and offset, respectively.

An angular distance metric was used to determine the nearest reference template. Each time bin in a test trial was classified as belonging to one of the following response categories: hexanol ON, hexanol OFF, benzaldehyde ON, benzaldehyde OFF or as an unclassified response. Those time bins that were not within a certain angular distance threshold (within 63° of the nearest reference template for Figs 5b and 8d and within 67° for Fig. 9b; Supplementary Fig. 9b) were categorized as unclassifiable responses. This threshold was chosen such that <10% of the ensemble neural activities in the pre-stimulus period were misclassified as being similar to the hexanol or benzaldehyde response templates.

**Information rate estimation.** We estimated the information content carried by the neural spike trains during ON and OFF response windows by computing the mutual information rate between odour stimulus and the neural response[61]. We used the 'direct method' approach by finding the difference between the total and conditional entropy rates of the responses[62,63].

$$I(S; R) = H_{total} - H_{noise} \qquad (2)$$

The total entropy rate ($H_{Total}$) was estimated using PN responses to five unique stimuli, and the conditional entropy rate ($H_{noise}$) was obtained from 25 repeated presentations of the same odorant (Supplementary Fig. 3). The unique stimuli used were hexanol 1%, 2-octanol 1%, hexanol 0.1%, 2-octanol 0.1% and the binary mixture of hexanol 1% and 2-octanol 1%.

**Cluster analysis for PN responses.** For clustering PN responses, we first binned each PN spiking response in 50 ms non-overlapping time bins (smoothed with a five-point average moving average filter). The PN responses over an 8 s period starting at the odour onset (160-dimensional vector) were then trial-averaged. All PNs with a statistically significant response (excitatory or inhibitory) were used for this cluster analysis. Responses recorded for the following four odorants were analysed: hex, 2oct, iaa and bzald (at 1% concentration v/v). PN responses were clustered such that the furthest pairwise distance between any two samples assigned to an individual cluster was minimized. A correlation metric was used as a measure of similarity:

$$Corr = \frac{\sum_{i=1}^{160}(x_i - \bar{x})(y_i - \bar{y})}{\sigma_x \sigma_y} \qquad (3)$$

where $x_i$ and $y_i$ are $i$th vector elements of two different PN response vectors, $\bar{x}$ and $\bar{y}$ denote the mean firing rate for each PN over the entire 8 s window, and $\sigma_x$, $\sigma_y$ represent the s.d. The optimum number of clusters required to represent the entire data set was chosen based on the mean-squared error (Supplementary Fig. 1a–c). Peak latency was calculated for the ON and OFF responses by finding the time bin with maximum firing rate after baseline subtraction (Supplementary Fig. 1d).

**Computational modelling of the locust antennal lobe.** Odour representation in the antenna was modelled with a repertoire of 50 ORNs. A subset of ORNs was activated by the stimulus, as shown in Supplementary Fig. 7a. Note that the sensory neuron response time constants for the rise, adaptation and fall phases were heterogeneous as found *in vivo*.

Next, the modelled sensory neuron responses (ORN responses) were input to a realistic computational model of the antennal lobe circuits with 50 excitatory projection neurons (PNs) and 25 inhibitory local neurons (LNs). Each PN was modelled as a regular spiking neuron and inhibitory local neuron as a fast-spiking neuron using a reduced Hodgkin–Huxley model[64].

$$\frac{dv}{dt} = 0.04v^2 + 5v + 140 - u + \left(I(t) - v_{thresh}^3 \times v_{memory}\right) \text{(fast variable)}$$
$$\frac{du}{dt} = a(bv - u) \qquad \text{(slow variable)} \qquad (4)$$

if $v = 30$ mV then $v = c$, $u = u + d$

PN model parameters: $a = 0.02$, $b = 0.2$, $c = -65$, $d = 8$.
LN model parameters: $a = 0.1$, $b = 0.2$, $c = -65$, $d = 2$.

$I$ is the total input to the each neuron from both sensory neurons, as well as summed contributions of other antennal lobe neurons. Note that the adaptive parameters ($v_{thresh}$ and $v_{memory}$) were limited to PNs only. The update rule for these two parameters is as follows:

$$\frac{dv_{thresh}}{dt} = inc \times \delta(t - t_s) - \frac{(4 + v_{thresh}(t))}{\tau_{thresh}} \qquad (5)$$

where $inc = 0.3$ and $\tau_{thresh} = 2,500$ ms for all PNs, $t_s$ is the time when the neuron last fired an action potential, and $\delta(t - t_s)$ is the Dirac delta function. Integration time step is 1 ms.

$$v_{thresh}(t) = \max \begin{cases} 0, \\ 2 \times \left[\frac{1 - \exp\left(-0.1 \times \int_{-\infty}^{t} h(t-s)v(s)ds\right)}{1 + \exp\left(-0.1 \times \int_{-\infty}^{t} h(t-s)v(s)ds\right)}\right] \end{cases} \qquad (6)$$

where $v(t)$ is the membrane potential of the neuron at time $t$ and h is a one-sided Gaussian kernel with s.d. uniformly distributed in the range (120, 320 ms).

*Model connectivity*. We modelled each PN to receive input from a single sensory neuron. LNs received input from nearly two-thirds of all sensory neurons. Further, since each LNs arborized extensively throughout the antennal lobe[65], each local neuron received excitatory input from roughly 30% of PNs, and provided feedback inhibition to ~30% of non-identical combination of PNs. Note that there were no excitatory lateral interactions between PNs or self-inhibition in the model. These connection probabilities and other network parameters including the type of synaptic currents were constrained based on estimates from locust antennal lobe circuits[65-68]. The connectivity matrix used in Fig. 7 is shown in Supplementary Fig. 7b–d.

The post-synaptic current generated by a pre-synaptic neuron $i$ following a spike at time $t$ was defined as follows:

$$\frac{dg(i, t)}{dt} = \frac{-g(i, t)}{\tau_{syn}} + z(i, t)$$
$$\frac{dz(i, t)}{dt} = \frac{-z(i, t)}{\tau_{syn}} + g_{norm} \cdot spk(i, t) \qquad (7)$$

where $z(.)$ and $g(.)$ are low pass filters of the form $\exp(-t/\tau_{syn})$ and $t \times \exp(-t/\tau_{syn})$, respectively, $\tau_{syn}$ is the synaptic time constant, $g_{norm}$ is the peak synaptic conductance (a constant), and spk($i$,$t$) marks the occurrence of a spike in neuron $i$ at time $t$. Synaptic parameters used were the following: peak synaptic conductance (excitatory synapse) = 0.1 nS, excitatory synapse response time constant = 5 ms, peak synaptic conductance (inhibitory synapse) = 0.3 nS, inhibitory synapse response time constant = 6 ms.

Therefore, the total synaptic current received by neuron $k$ from all other neurons in the network is given by:

$$I_{syn}(k, t) = \sum_{\forall i \neq k} C_{ik} \cdot g(i, t) \qquad (8)$$

where $C$ is the recurrent connectivity matrix (refer Supplementary Fig. 7d).

The total input to the neuron $k$ taking into account both sensory input and synaptic inputs received through recurrent connections can be written as follows:

$$I(k, t) = W * ORN(t) + I_{syn}(k, t) \qquad (9)$$

where $W$ is the input connection linking ORNs with PNs and LNs (refer Supplementary Fig. 7b,c respectively), ORN($t$) is the input vector representing the sensory neuron activity at time $t$ (refer Supplementary Fig. 7a).

*LFP and sliding-window cross-correlograms*. The LFP in the model was computed as the sum of PN membrane potential fluctuations (filtered between 5 and 55 Hz). The pairwise cross-correlations were obtained by averaging LN membrane potential fluctuations in 500 ms time windows (98% overlap between consecutive time segments) and comparing them with LFP during the matching time segment (Supplementary Fig. 7f).

**Bifurcation analysis.** The inhibition was regularized by multiplying a scaling constant (between 0 and 1) to the synaptic weights from LNs to PNs, and the bi-directional adaptation was regularized by multiplying a scaling constant (between 0 and 1) to the update step $\Delta v_{thresh}$. For every combination of the inhibition and the bi-directional adaptation, we calculated the total LFP power in the 5–55 Hz frequency range, and the angle between the ensemble ON and OFF responses. The angle between ON and OFF responses was obtained by first binning the data into 50 ms time bins and calculating the response similarity between mean ON response vector and the mean OFF response vector (Fig. 7).

**Behaviour experiments.** Behavioral experiments were performed on locusts of either sex that were starved for a day before their use in the appetitive-conditioning assay. The same protocol as described elsewhere[8] was used. Briefly, locusts were immobilized, eyes closed using a black tape, and palps painted using a zero-volatile-organic-chemical green paint (Valspar ultra). The training sessions began an hour after the palps were painted.

Hexanol (hex) was used as the conditioned stimulus for all experiments and wheat grass was used as the unconditioned stimulus. Odour delivery setup was identical to that of electrophysiology experiments. A video camera (Microsoft webcam) was used to capture the palp movement at 25 or 30 frames per second. Odour delivery and video tracking data acquisition was controlled in an automated fashion using a custom written Labview program.

During each training trial, conditioned stimulus was presented for 10 s. Food reward was given 4–5 s after the onset of the conditioned stimulus.

The training phase included a total of six training trials with a ten minutes interval between successive training trials. Only locusts that ate wheat grass in four out of the six training trials were retained for the testing phase (∼88% of the locusts used).

In one set of experiments (results reported in Figs 5 and 8), the testing phase included a total of five trials. The order in which different stimuli were presented was pseudo randomized. The inter-trial delay was set to 20 min. Locust PORs were collected for hexanol pulses of three different durations (0.75, 4 and 10 s duration), a benzaldehyde pulse of 4 s duration; and an overlapping sequence of hexanol and benzaldehyde. As in our physiology experiments, the overlapped presentation had a 4 s pulse of hexanol followed by a 4 s benzaldehyde pulse with 0.5 s overlap between the pulses.

In the second set of experiments (results shown in Fig. 9, Supplementary Fig. 9), the testing phase included a total of four trials. Locust PORs were recorded for a hexanol 6 s pulse, a 2 s ON–2 s OFF–2 s ON pulsatile stimulation of hexanol, a synchronous binary mixture of hexanol and benzaldehyde, and a benzaldehyde pulse of 6 s duration.

Locusts were kept on a 12 h day–12 h night cycle (0700 hours to 1900 hours day). All behavioral experiments were performed between 0900 hours and 1500 hours.

**Palp-tracking algorithm.** Maxillary palp movements were analysed offline using a custom written Matlab program. The goal of the processing was to provide contrast to enable tracking of the palps that were painted with a non-odorous green paint.

Each video file was converted into a time series of RGB colour frames. For each frame, the grayscale image was subtracted from the green channel of that frame. Then, a 2-D averaging filter was applied to remove extraneous pixels in the background. Pixel intensities of the filtered image were trimmed to range between the maximum intensity value and a manually set minimum threshold value. The filtered image was subsequently remapped onto a wider intensity range (0–255 UINT8) to allow for more robust tracking of palps across frames and across videos. The image was then converted to binary with a manually set threshold to generate a matrix, where the painted palp segments had a value of 1 (HIGH). A series of adjustments were then performed on the image to ensure proper tracking of each palp. If HIGH non-palp regions were present, they were removed by comparing their corresponding UINT8 intensity values to those of the palp regions. If the locations of one or both palps were blocked due to movement of the antenna or other focusing issues, the palp positions were estimated using their positions in previous frames. If the palps were overlapping (resulting in only one HIGH region), they were split at the centroid of the region. After these adjustments, the centroids of the two maxillary palp segments were located and were tracked across video frames. A rectangular region of interest (ROI) was created based on the centroid positions for each frame to reduce the background size (and therefore processing time) in the subsequent frame. For each video analysed in this fashion, the tracking results were manually inspected and the tracking was rerun with adjusted parameters if necessary. Such adjustable parameters included: the padding of the ROI around palps, the minimum threshold for intensity mapping, and the splitting direction for overlapping palps. Data from all trained locusts were included in our behavioral response analyses shown in Fig. 8 (hex-bzald overlap). For the POR responses shown in Fig. 9 (hex 6 s and pulsatile stimulation), 3-out-of-30 locust POR responses were excluded from analyses as rapid antenna movements interfered with palp tracking.

**Modelling of behavioral results.** To predict behavioral responses from ensemble neural response data, we used four simple models.

*ON model.* The ensemble neural activity in a particular time bin $x(t)$ became the input to the model. The probability of pattern-match between $x(t)$ with the hexanol ON response template was computed as in the classification analysis (red curves from Fig. 5c). The pattern-match probabilities were thresholded and adjusted for gain to predict behavioral response adjustment $\Delta y(t)$. The current behavioral response was just a simple linear sum of the behavioral response in the previous time bin $y(t-1)$ and the predicted adjustment for the current time bin $\Delta y(t)$. The entire model can be summarized using the following set of equations:

$$\Delta y_{\text{Open}}(t) = \varphi(\alpha_{\text{ON}}(t)) \times g_1, \varphi(\cdot) = \begin{cases} 1 & \alpha_{\text{ON}}(t) \geq \text{thresh}_{\text{ON}} \\ 0 & \alpha_{\text{ON}}(t) < \text{thresh}_{\text{ON}} \end{cases}$$

$$\Delta y_{\text{close}}(t) = \text{Not}(\alpha_{\text{ON}}(t)) \times g_2 \times y(t-1), \quad \text{Not}(\cdot) = \begin{cases} 1 & \alpha_{\text{ON}}(t) = 0 \\ 0 & \alpha_{\text{ON}}(t) > 0 \end{cases} \quad (10)$$

$$\Delta y(t) = \Delta y_{\text{Open}}(t) + \Delta y_{\text{close}}(t)$$

where $\alpha_{\text{ON}}(t)$ is the probability of pattern-match of the average ensemble activity in a given time bin $(x(t))$ with the hexanol ON template (same as shown in Fig. 5c; red curves), $\varphi(\cdot)$ indicates the nonlinear thresholding function, and $g_1 = 1$ indicates the gain. $\text{Not}(\cdot)$ is the NOT gate function, $\text{thresh}_{\text{ON}} = 0.6$, $g_2 = -0.05$ is another gain, and $y(t-1)$ is the behaviour output at the previous time point.

*ON–OFF model.* In this model, palp-opening response was solely determined based on the degree of pattern-match with the hexanol ON template (Fig. 5c; red trace), and the palp-closing was solely determined based on the degree of

pattern-match with the hexanol OFF-template (Fig. 5c; blue trace). This can be summarized as follows:

$$\Delta y_{\text{Open}}(t) = \varphi_{\text{ON}}(\alpha_{\text{ON}}(t)) \times g_{\text{ON}}, \quad \varphi_{\text{ON}}(\cdot) = \begin{cases} 1 & \alpha_{\text{ON}}(t) \geq \text{thresh}_{\text{ON}} \\ 0 & \alpha_{\text{ON}}(t) < \text{thresh}_{\text{ON}} \end{cases}$$

$$\Delta y_{\text{close}}(t) = \varphi_{\text{OFF}}(\alpha_{\text{OFF}}(t)) \times g_{\text{OFF}}, \quad \varphi_{\text{OFF}}(\cdot) = \begin{cases} 1 & \alpha_{\text{OFF}}(t) > \text{thresh}_{\text{OFF}} \\ 0 & \alpha_{\text{OFF}}(t) = \text{thresh}_{\text{OFF}} \end{cases}$$

(11)

where $\alpha_{\text{OFF}}(t)$ is the probability of pattern-match of the average ensemble activity in a given time bin $(x(t))$ with the hexanol OFF template (same as shown in Fig. 5c; blue curves), $\text{thresh}_{\text{ON}} = 0.6$, $\text{thresh}_{\text{OFF}} = 0$, $g_{\text{ON}} = 1$ and $g_{\text{OFF}} = -0.7$. All other variables and constants are the same as in equation (10).

For Fig. 9, we recorded and used a new set of neural and behavioral data (neural *dataset4*: hex and bzald). The ON–OFF model parameters in this case were obtained by fitting the predictions to the POR evoked by 6 s hexanol puff ($\text{thresh}_{\text{ON}} = 0.36$ and $g_{\text{OFF}} = -0.55$).

*Active ON model.* In this model, a pattern-match with ON template drives both opening and closing of the palps (Supplementary Fig. 8a). The model can be summarized using the equation below.

$$\Delta y_{\text{Open}}(t) = \varphi(\alpha_{\text{ON}}(t)) \times g, \varphi(\cdot) = \begin{cases} 1, & \alpha_{\text{ON}}(t) \geq 0.6 \\ -1, & \alpha_{\text{ON}}(t) < 0.1 \\ 0, & \text{otherwise} \end{cases} \quad (12)$$

Similar to equation (10), $\varphi(\cdot)$ indicates the nonlinear thresholding function, and $g$ indicates the gain. Note that $g$ was assigned two different values depending on whether palp was opening or closing, that is, if $\varphi(\cdot) = 1$ then $g = 1.0$, alternately if $\varphi(\cdot) = -1$ then $g = 300$.

*Passive ON–OFF model.* In this model, a pattern-match with the ON template is sufficient to initiate and sustain the POR responses. However, the mismatch with both ON and OFF response templates triggered the palps to close. The following equations summarize the model:

$$\Delta y_{\text{open}}(t) = \varphi(\alpha_{\text{ON}}(t)) \times g_1 \quad \varphi(\cdot) = \begin{cases} 1 & \alpha_{\text{ON}}(t) \geq 0.6 \\ 0 & \text{otherwise} \end{cases}$$

$$\Delta y_{\text{close}}(t) = (\alpha_{\text{ON}}(t), \alpha_{\text{OFF}}(t)) \times g_2 \times y(t-1) \quad (13)$$

$$\text{Not}(\cdot) = \begin{cases} 1 & \alpha_{\text{ON}}(t) = 0, \alpha_{\text{OFF}}(t) = 0 \\ 0 & \text{otherwise} \end{cases}$$

Here, $g_1 = 1$, $\alpha_{\text{ON}}(t)$ and $\alpha_{\text{OFF}}(t)$ are the probabilities of pattern-match with the ON and the OFF template, respectively, $g_2 = -0.05$, and $y(t-1)$ is the POR at the previous time point.

We fit the models and selected their parameters using the behavioral data observed for hexanol 4 s exposures. The models were tested based on their predictions to behavioral responses to hexanol pulses of other durations and to the overlapping presentation of hex-bzald (Fig. 8c,e, Supplementary Fig. 8b). Since the models were designed primarily to predict the palp opening and closing dynamics, we rescaled the amplitude of the predicted responses to fit the experimentally observed peak PORs values for each condition. Also, to match the sampling rate of the behavioral data (25 frames per second) and classification probabilities (50 ms time bins), both neural pattern-match and PORs were re-binned using 200 ms time bins.

To quantitatively compare the performance of different models, we computed the mean squared errors (MSE) between predicted and actual rise time and fall time constants for palp-opening and palp-closing responses (Supplementary Fig. 8c). Rise time was defined as the time taken for the median palp-opening response to reach 50% of the peak palp separation distance from odour onset. Similarly, fall time was defined as the time taken for the palp closing responses to reduce to 50% of the palp separation distance after the conditioned stimulus was terminated. The MSE of prediction was computed as follows:

$$\text{MSE} = \frac{1}{4} \sum_{i=1}^{4} (Y_i - X_i)^2 \quad (14)$$

where **Y** represents the set of values predicted by the model and X is the measured POR responses. Responses to three different durations of hexanol presentation and to the hexanol-benzaldehyde odour sequence were used for computing MSE.

**Granger causality test.** We examined whether the results obtained from our classification analysis (physiological data) were Granger causal with the behavioral PORs for the hex-0.5 s overlap-bzald stimulus sequence (Fig. 8d,e). For this time series analysis, we first combined the pattern match probabilities with ON and OFF response templates of hexanol (conditioned stimulus) as follows:

$$\alpha_{\text{ON-OFF}}(t) = \alpha_{\text{ON}}(t) - \alpha_{\text{OFF}}(t) \quad (15)$$

We used $\alpha_{\text{ON-OFF}}$ as one time series vector and the behavioral POR as the second. In order to have the same sample size for the behavioral POR and the classification template match probability vector, we generated classification results with 200 ms temporal resolution. The significance level was set to 5% and the maximum lag between the two time series was set to 10 samples (that is, 2 s).

**Auditory electrophysiology.** Neural responses were collected from the primary auditory cortex (A1) of two marmoset monkeys (*Callithrix jacchus*). Details of the recording procedure were consistent with previous studies[69,70]. A one-time surgery was performed to implant a head cap on the animal for head fixation. During recording, the animal sat comfortably in a custom-made primate chair and passively listened to the sound stimuli. The animal's head was held by a fixation bar. Craniotomy holes were drilled over the putative A1 area identified through the location of the lateral sulcus and measurements according to a standard marmoset brain atlas[71]. The location of A1 was further confirmed through the topological mapping of the neurons' best frequencies. Single tungsten electrodes ($\sim 5\,M\Omega$ impedance at 1 kHz, FHC) were used to record neural activities from the craniotomy holes. The recorded extracellular signals were amplified 1,000 times and filtered between 0.1 and 5,000 Hz (AM Systems 1800). They were then fed into an online template-based spike sorting software (Alpha-Omega). Multi and single units were determined based on the visual inspection of action potential waveforms and quantification of inter-spike interval distribution. Units with $> 0.45\%$ spikes of $< 1\,ms$ inter-spike-interval were labelled as multi-units. Only single units were included in the analysis (171 single units in total).

The recordings were carried out in a double wall sound proof booth (IAC 120a-3). A free field speaker (B&W 601S3) was placed exactly 1 m in front of the animal to deliver sound stimuli. A large variety of sounds were typically used to search for neurons to avoid bias in neuron selection, including random spectrum stimuli[72], pure tones, amplitude or frequency modulated tones, band pass or band stop noises, clicks and marmoset vocalizations. Neurons were included for further test and analysis, as long as they responded to any of the searching stimuli.

Experimental sound stimuli were pure tones of 1, 2, 4, 8, 16, 32 kHz, each presented at four sound levels (85, 55, 25, $-5\,dB$ SPL, sound pressure level). Each tone was 0.5 s long with 5 ms onset and offset ramp. Inter-trial-interval was at least 1 s. Recordings of each trial included 0.3 s before and 0.7 s after sound presentation. Stimuli were presented in pseudo-random order, with 20 trials collected for each stimulus. Only data for the highest intensity (85 dB SPL) tones was used for analysis.

**Sorting of A1 responses.** A similar procedure used for sorting PN responses was used for sorting A1 responses (Fig. 10c–h, left panel). Specifically, a peak ON response (Peak ON) and a peak OFF response (Peak OFF) were obtained for each neuron as the maximum firing rate in non-overlapping 25 ms bins during either the 0.5 s sound presentation duration or the 0.5 s duration after sound termination. The larger of the two values were then defined as the peak firing rate during the entire duration (Peak total). Responsive neurons for each sound as determined below were sorted by the difference between the peak ON response and the peak OFF response, normalized by the total peak firing rate ($\frac{Peak\ ON}{Peak\ total} - \frac{Peak\ OFF}{Peak\ total}$). Nonresponsive neurons were plotted in random order under responsive neurons.

To determine neuron responsiveness, neural responses of 1 s duration starting from sound onset were used, including 0.5 s sound presentation, and 0.5 s window after sound termination. Excitatory and inhibitory responses were determined separately. Specifically, satisfaction of two conditions was required to qualify an excitatory response. (i) mean response in at least one time bins (25 ms) during the entire 1 s window must exceed 2 s.d. of the mean baseline activity[70]. (ii) the criterion in (i) must be met by at least 4 out of the 20 individual trials.

A significant inhibitory response was determined when the overall mean response during either the ON or OFF response epochs was less than the mean baseline activity, and no time bin within that epoch could exceed the mean baseline activity. In addition, these criteria must be satisfied in at least 4 of the trials.

**Dimensionality reduction analysis for A1 responses.** The same methods used for visualizing olfactory neuron activities were used for A1 neuron responses calculated in 30 ms Gaussian windows with 5 ms increments. Since the time bins overlapped, no further smoothing of the generated trajectories was done. Trajectories included responses of 1 s duration starting from sound onset (0.5 s ON responses during sound presentation, 0.5 s OFF responses after sound termination) (Fig. 10c–h, right panels).

**Angle between the mean ON and OFF A1 neuronal responses.** The same method used for comparing PN responses was also utilized to quantify the angle between the mean ON and OFF responses in A1 neurons. ON response was averaged over the 0.5 s sound presentation duration. OFF response was averaged over the 0.5 s duration after sound termination. Baseline activity derived from the 0.3 s recording before sound onset was subtracted from both the ON and OFF response vectors. A bin size of 0.25 s was also used to confirm the observed relationship between the two response components (Fig. 10i).

**Justification of statistical tests.** All statistical significance tests done in the manuscript were two-sided. Bonferroni-corrected *P* values were used in case of multiple comparisons. No statistical methods were used to predetermine sample sizes, but our sample sizes are similar to those reported in previous publications in the field.

For the paired *t*-tests, normality of the dataset was confirmed using the Jarque–Bera test. The equal variance assumption was tested using the Levene's test

(Fig. 3e,f). The confidence level was set to 0.05. Wilcoxon signed-rank test is a non-parametric test for comparing the population median responses of matched samples. This test was used to detect when a significant decrease in palp-closing responses occurred (Fig. 8e,f).

For the two sample Kolmogorov–Smirnov test, we used a significance level of 5% to check if the two vectors are from the same distribution (Supplementary Figs 1d,3).

**Data availability.** The data used in this study can be made available on reasonable request to the authors.

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

## Acknowledgements

We thank Haney Seth and Maxim Bazhenov (University of California, Riverside), Shantanu Chakrabartty (Michigan State University) and Shinung Ching (Washington University in St. Louis), and members of Raman and Barbour Lab for feedback on the manuscript. This research was supported by a McDonnell Center for Cellular and Molecular Neurobiology fellowship to D.S., an Office of Naval Research grant (N00014-12-1-0089) and a NSF CAREER grant (#1453022) to B.R., and NIH grant DC-009215 to D.L.B. Support from Center for Biological and Systems Engineering facilitated open-access publication of this manuscript.

## Author contributions

B.R. conceived the study. D.S. and B.R. designed the olfaction experiments. C.L. and B.R. designed the analysis methods. Z.C. and B.R. developed the computational model. W.S. and D.L.B. designed the audition studies. D.S., C.L., B.R. and W.S. performed the experiments and the analysis. W.P. performed the behavioral experiments and their analysis. S.N., A.C., E.A., R.L. performed portions of additional experiments and analyses that were necessary to address the reviewers' comments. D.S. wrote the methods section and figure captions for olfaction experiments. W.S. and D.L.B. wrote the sections on audition experiments and their methods. B.R. wrote the paper, incorporating inputs and comments from all authors. B.R. and D.L.B. provided overall supervision.

## Additional information

**Competing interests:** The authors declare no competing financial interests.

