## [Peer Review File · Nature Communications]

Reviewers' Comments:

Reviewer #1 (Remarks to the Author):

General Comments:

In this study, the authors attempt to investigate the importance of the temporal patterning of population neural activities. They conducted extracellular recordings in the locust antennal lobe in response to 4 different odorants and identified near orthogonal population coding of the stimulus-on and stimulus-off epochs from about 80 projection neurons (PNs). The on and off responses to the same odorant appear more different than the on responses between different concentrations of the same odorant and different odorants of the same concentration, thus making it possible to classify the population responses based on angular similarity of the population response vectors. The authors also conducted intracellular recordings from 5 inhibitory local interneurons (LNs) and 5 PNs, together with local field potential recordings in the mushroom body, finding no extended LN response and no oscillatory activity beyond the stimulus period. The authors next characterized the palp opening responses in locusts trained to associate an odor (CS+) with a food reward (US) and attempted to differentiate 4 different models based on responses to a partially overlap sequence of CS+ odor and CS- odor. Finally, the authors included recordings in the marmoset auditory cortex and showed similar on and off responses to sound stimuli.

Overall, the work involved is extensive, the references are thorough, the main texts are clearly written and the statistical methods are sound. However, in both the abstract and the conclusions, the authors appear to have over-interpreted the correlations observed and concluded prematurely the role of the local inhibitory neurons and the significance of the behavioral responses to the partially overlapping sequence of odors. Unfortunately, the work required to definitively address them appear very difficult to be conducted in locusts and may be better done in a genetically tractable system. Having the marmoset data is nice, but its inclusion does not seem to enhance the main point of the study beyond phenomenological similarity.

As stated in the discussion, responses to stimuli onset and offset have been reported across a variety of sensory stimuli and organisms. Therefore this study represents an incremental expansion rather than a conceptual advance.

Specific Comments:

1. The "millisecond" in the beginning of the 6th row in Introduction should be plural.

2. Figure 1a: Maybe should also quantify the responses as normalized to the pre-stimulus period? For example, the PN2 "peak" response to 1% 2oct on is not really an "on" response, which should actually be inhibition.
3. Figure 1b: Left: What's the scale for the mean firing rate? Please include in the figure. Right: the "spokes" are distracting. Is the steady state fixed point same as the origin or just masked by the spokes? Maybe code time as the color intensities of the trajectories?
4. Figure 1f: Does the 2s time window start at odor onset and offset or in the middle?
5. Figure 2: The observation that the most difference is seen between the on and off periods of the same odorant might just be due to the inhibition during the odor on period preceding the prominent off responses. Does the observation generalize to the odorants other than the 2 shown?
6. Figure 5: Why use 1s odor delivery instead of 4s as in the previous PN recordings? The absence of the observation of prolonged LN response to odor after delivery is not proof that no LN has prolonged response, given the small number of LNs recorded. This is at best just correlation from a limited sample size and does not justify the causality implied in the title and abstract. On the other hand, could the absence of LN odor-off response be explained by the absence of activity from ORNs rather than the disengagement from the PNs?
7. Figure 6: The modeling exercise only suggests a potential mechanism, which has not been proven to be actually true in this study.
8. Figure 7: The authors used a CS- odor, benzaldehyde, partially overlapping with the CS+ odor, hexanol, to interfere near the end of the hex on response and the start of the hex off response. Here they appear to be over-interpreting the data. As shown in the bottom panel of Figure 7d, the population activity induced by the overlapping odors is quite complex, with the bzald-on peak pushing apart the hex-on and hex-off epochs, and the hex-off epoch overlapping with the second peak of the bzald-on epoch, finally with both hex-off and bzald-off epochs overlapping. The POR result for the overlapped odor sequence is not as clean cut as the authors stated either. The POR during hex-off/bzald-on seems clearly higher than both the baseline and the hex alone case, while the authors interpreted the first time point of the dip in palp separation as the start of palp closing without taking into account the sustained "half-opening" of the palps. Moreover, the seemingly same delay between the rise of hex-off probability curve and the first dip in palp separation does not warrant the strong statement of the stimulus-evoked off responses being REQUIRED for behavioral reset, which can only be addressed by somehow eliminating the odor off responses and seems impossible to do in the current system.
9. Figure 8: There is no data on any local inhibitory interneurons so it is not clear whether the underlying mechanism is similar to that of the locust PNs.

10. Supplementary Figure 1: There are a lot more ORNs in the antennae than the PNs in the antennal lobe. It is fair to compare data from around 20 ORNs to that from 80 PNs. Maybe the absence of the odor off response is due to the small sample size for ORNs.

Reviewer #2 (Remarks to the Author):

This manuscript analyzes some interesting features of ON and OFF responses and considers the important question of what makes these patterns of neural activity differ, which is especially important in contexts in which behavior is driven at short latency and so can be completely explained by no responses. Unfortunately many of the data are presented in a non-quantitative way, despite considerable quantitative expertise of the PI and the conclusions are not well-supported by the data. Mechanisms to explain the differences in OFF and ON responses are insufficiently explored as well.

The abstract and introduction of the paper make rather strong claims about the connection to behavior, which I find somewhat overstated. The authors identify a correlate of behavior, but there is nothing in the off responses that specifically have been identified as encoding features of the behavior, nor is the mechanism by which off responses differ in their ability to recruit inhibition etc well explained in the authors' results.

The claim that in general ON responses for different odors are more similar than ON and OFF responses for the same odor should be further analyzed. Can "typical" features of OFF responses be identified? Perhaps an analysis of correlation would provide some insight? If so, what are they and how may they be explained mechanistically? Related to this is the observation that spike-field synchrony is reduced during the OFF responses even when avg firing rate is relatively high. The authors suggest that this reflects differential engagement of inhibition, but evidence in favor of this is not strong. The authors' interpretation of these data to state that "neural encoding formats vary significantly" seems overly loaded and cumbersome. Without a consensus on what a neural encoding format means, this statement is not very meaningful.

For figure 2, some clearer, more intuitive way of displaying the data to emphasize the authors' point would be very helpful. I would recommend showing spike rasters but showing them in order of the change in firing rate for ON responses for one odor. This would facilitate the readers' understanding of the analysis. Moreover, some description of the approach to the analysis of the data and the degree of difference between ON and OFF responses vs different odor responses should be included.

Generally more description of data means and significance should be included in the text. Statements like: "We found that in general, almost all PNs that were activated during stimulus exposure period were inhibited following stimulus termination with the firing activity reaching

below baseline levels.” are extremely unsatisfying. Is this a significant effect or not? What does “almost all” mean?

Some descriptive statistics would be helpful. What fraction of cells show increased firing rates during ON or OFF responses? What is the overall distribution of these responses? Simply showing mean changes in firing rate may disguise much of the interesting data.

How significant and/or surprising is this observation that the ON and OFF responses are approximately orthogonal? (pg 2) For example, if the authors took ON responses from one set of cells and appended them to OFF responses of a different set and then performed PCA, would they still observe this orthogonal relationship? Or not?

I expected the section labeled “Information Content Analysis” to include some information theoretic analysis of the population activity such as a quantification of total and conditional entropy. How many bits of information are available from the ON vs the OFF responses? Authors discuss anecdotally the information content of off responses as in “In sum, these results suggest that OFF responses are as consistent as the ON responses, and they actively convey information about the termination of a particular stimulus at a specific intensity.” Quantitative assessment of these claims is critical to include in the body of the MS.

The features of the model that are important for mediating the kinds of responses observed are not clearly tested, lessening the impact of the model. Some mechanistic predictions would make the model of greater interest and strengthen the manuscript as a whole.

Reviewer #3 (Remarks to the Author):

Saha and colleagues present an important new set of observations about the representation of odor onset and offset in the insect antennal lobe, taking advantage of the electrophysiological accessibility of the locust antennal lobe to carefully measure circuit-level properties. The core result is that the patterned activation of projection neurons by removal of neurons is a key component of ensemble encoding of behaviorally relevant sensory information.

All of the experiments are beautifully designed and presented. I am persuaded by all of the arguments, and I have only minor comments that might help make the paper more understandable to its readership.

My understanding is that the "OFF" responses of the PNs is intimately connected to and shaped by the inhibitory circuit. Post-inhibitory rebound is not a new phenomenon, and understanding precisely how it plays a role in the antennal lobe is a significant contribution. Towards this end, it would be good to spend more time describing the model and simulation. A diagram of the model with pertinent synaptic connections would be helpful, as it would clarify the fact that ORNs activate LNs, LNs inhibit ORNs and PNs, and PNs activate LNs. Illustrations of key

dynamical features within the diagram would also help. For example, it would help to convey the "bidirectional adaptive threshold mechanism" that has been posited to occur in the projection neurons, which is apparently fundamental to bringing the ON responses in line with the OFF responses. As far as I can tell, that such a mechanism exists is a prediction of the model that remains to be tested.

It is unclear to me that the "bidirectional adaptive threshold mechanism" is really what generates the orthogonal ensemble activities, although it would certainly promote it. In the recent Nagel and Wilson (2016) paper in the Journal of Neuroscience, local neurons in *Drosophila* were shown to have both "ON" and "OFF" responses, and a similar PCA analysis showed that the first two principal components separately described these responses. If their LN dynamics were thus plotted in the same way as the locust PN data, I would also expect orthogonal representations.

The authors alternately refer to "angular similarity" and "angular distance" when referring to the same thing. Be consistent.

The authors design a very clever odor associated learning experiment to test the behavioral relevance of the "OFF" response. They train the animal with a certain odor until the animal initiates feeding behavior when the odor is presented and terminates feeding when the odor is removed. The key experiment is what happens when adding an untrained odor to create a mixture before the trained odor is removed. Animals terminate the response when the trained odor is removed, and not when the "template match" to the trained odor is erased. The underlying assumption, I think, is that there is a single classifier at work. But what would happen if there are multiple classifiers, each sensitive to different subsets of PNs? I think this could be arranged to generate the same result, allowing the template match to the trained odor to be largely preserved in one classifier even when a new odor is added to create the mixture. One might test if there is only one classifier at work, by delivering and removing the mixture all at once. If the animal fails to respond, then the animal would not be recognizing the presence of the trained odor within the mixture by a more specialized classifier. If it does respond, then it would be picking out the addition and removal of the trained odor irrespective of the distracting odor.

Reviewer #4 (Remarks to the Author):

In the manuscript "Engaging and disengaging recurrent inhibition mediates sensing and unsensing of a

sensory stimulus" is an interesting study. In this study, authors make three major points (in my opinion) – 1) That odor ON and OFF activate different populations of neurons, 2) That, in the locust, this orthogonality might result from different degrees of recurrent inhibition engaged by odor-ON and odor-OFF. 3) That neurons that mediate ON response result in behavior initiation. More interestingly, the neurons that are activated during OFF are important for behavior

termination. I think that the importance of the manuscript lies in making the community think about behavior termination as an active process that needs active neurons to turn it off. In that light, the manuscript makes an important contribution. The experiments are well-executed and the analysis is insightful.

I do have 3 critiques. The first critique is aimed at further strengthening the evidence for the behavioral experiment. The other two critiques mostly relate to streamlining the manuscript.

1. RELATION BETWEEN NEURAL RESPONSE AND BEHAVIOR: The authors have done a reasonable job at convincing me of their result that the neurons that are turned OFF drive response termination. But, I am not totally convinced. The following things might help: i) I have several general questions about the behavior itself. I am quite surprised that up to 10 s, the locust keeps its palp open. Does it make sense that the locust keeps its palps open for 10s? I imagine that if one kept the odor on for even longer, it would eventually shut its palp. Have the authors explored this issue? Also, how does this result compare to other classical conditioning paradigms?

Another general question is that there is clearly a really long tail to the behavior. That is, the palp separation is greater even 10 seconds after the odor is turned off (Figure 7C). I doubt that that long tail can be explained by the dynamics of the neurons in the antennal lobe. And that's OK. We all know that there are several steps between the antennal lobe and behavior. I would suggest finding a method to subtract the long tail. The exact method would depend on whether this is seen before conditioning or only after conditioning. This is important because I think it will have a great effect on the fits.

ii) It appears that the ON model is a better fit to the data in Figure 1C.

iii) I like the experiment with benzaldehyde. It is quite creative. But, I am not sure I fully understand it. Why does the addition of benzaldehyde change the hex-off. I am also wondering why the authors did not do the much simpler experiment of giving several pulses of hexanol. Compare 4s pulse of hexanol to 2s-ON-2s-off-2s-ON.

2. INTRODUCTION AND GENERAL PITCH: I found the introduction to be too broad and all encompassing and found myself not agreeing with many of the statements. I suggest that the authors should start with the simple premise that in most systems there are neurons that turn ON when the stimulus is ON and others that turn ON when the stimulus is OFF and that they are going to explore that motif.

3. MECHANISM FOR ORTHOGONALITY: I am a little confused as to what the authors think is critical for orthogonality. They mention that "bidirectional adaptive threshold" was important. I think that that is all one needs to generate orthogonality. Essentially PNs downstream to ORNs that are inhibited will be turned ON when the odor is turned off. This occurs in every sensory

system, and this is all you need. In that sense, the evidence for recurrent inhibition is cool but specific to locust. The phenomenon, itself, is likely to be ubiquitous.

Summary of changes made to address Reviewers' concerns:

1. New electrophysiology experiments:

We have made additional extracellular recordings from 85 PNs characterizing their responses to the following cues:

1. hexanol (2s ON – 2s OFF – 2s ON; i.e. pulsatile presentations of hexanol. This experiment was recommended by Reviewer 4),
2. hexanol (6s; for response comparison with pulsatile cue)
3. mixture of hexanol and benzaldehyde (recommended by Reviewer 3)

In addition, we have also done electrophysiology experiments to illustrate that fast GABAergic inhibition is important for generating oscillatory local field potentials (LFP) as observed during odor ON responses. Blocking inhibition pharmacologically using picrotoxin reversibly abolished oscillatory LFPs and thereby eliminated the differences observed in the LFP during odor ON and OFF periods (refer to **Supplementary Fig. 7** in the revised manuscript).

These additional experiments nicely support our conclusions as explained below in the point-by-point responses to reviewers' concerns.

2. New behavioral experiments:

As recommended by Reviewers 3 and 4, we have now examined the responses of locusts conditioned with hexanol to the following cues:

1. hexanol (2s ON – 2s OFF – 2s ON),
2. hexanol (6s)
3. mixture of hexanol and benzaldehyde

The multiphasic palp opening responses (PORs) observed due to the pulsatile application of hexanol (correlating with the ON-OFF-ON-OFF neural activities), and the reduced responses to the mixture of hexanol and benzaldehyde responses (reflecting changes in neural pattern match with hexanol ON responses) lend further support to our conclusions. We thank the reviewers for making these important experimental suggestions.

3. Expansion of the modeling work:

We have made simulations where we have systematically shown the functional role of each of the following components in the model:

- a. Feed-forward inputs to local neurons
- b. Feedback inhibition from local neurons onto PNs
- c. Bidirectional modulation of PN excitability

We believe that these simulations and the new experimental data abolishing LFP oscillations with picrotoxin injection significantly improve the contributions of our modeling work.

4. New analyses performed:

There were several interesting suggestions regarding additional analyses that have all been done now. These include:

- Additional dimensionality reduction analyses to show the differences between ON and OFF responses across odorants,
- Systematic characterization of the evolution of neural trajectories over time,
- Self- and cross-correlation analyses that compare similarity between population neural vectors obtained at different points in time,
- Information theoretic analyses characterizing the amount of information obtained from each projection neuron during the ON and OFF epochs (as suggested by Reviewer2).

- Cluster analyses to reveal the response motifs obtained during ON and OFF response epochs
- A Granger causality analyses to show that POR responses can be better predicted with neural pattern matches observed during the ON and OFF epochs than those made using POR response histories alone.

In sum, we believe that we have addressed each and every concern raised by the reviewers.

Reviewer #1 (Remarks to the Author):

General Comments:

In this study, the authors attempt to investigate the importance of the temporal patterning of population neural activities. They conducted extracellular recordings in the locust antennal lobe in response to 4 different odorants and identified near orthogonal population coding of the stimulus-on and stimulus-off epochs from about 80 projection neurons (PNs). The on and off responses to the same odorant appear more different than the on responses between different concentrations of the same odorant and different odorants of the same concentration, thus making it possible to classify the population responses based on angular similarity of the population response vectors. The authors also conducted intracellular recordings from 5 inhibitory local interneurons (LNs) and 5 PNs, together with local field potential recordings in the mushroom body, finding no extended LN response and no oscillatory activity beyond the stimulus period. The authors next characterized the palp opening responses in locusts trained to associate an odor (CS+) with a food reward (US) and attempted to differentiate 4 different models based on responses to a partially overlap sequence of CS+ odor and CS- odor. Finally, the authors included recordings in the marmoset auditory cortex and showed similar on and off responses to sound stimuli. Overall, the work involved is extensive, the references are thorough, the main texts are clearly written and the statistical methods are sound. However, in both the abstract and the conclusions, the authors appear to have over-interpreted the correlations observed and concluded prematurely the role of the local inhibitory neurons and the significance of the behavioral responses to the partially overlapping sequence of odors. Unfortunately, the work required to definitively address them appear very difficult to be conducted in locusts and may be better done in a genetically tractable system. Having the marmoset data is nice, but its inclusion does not seem to enhance the main point of the study beyond phenomenological similarity. As stated in the discussion, responses to stimuli onset and offset have been reported across a variety of sensory stimuli and organisms. Therefore, this study represents an incremental expansion rather than a conceptual advance.

We thank the reviewer for finding our work extensive and clearly written. As we explain in this document, we believe we did not over-interpret the role of inhibition, or the relevance of odor-evoked OFF responses to behavior. In fact, several suggestions made by this reviewer and others have helped us to amplify our points and further support our conclusions.

We would like to emphasize that although ON and OFF responses have been reported, we found that distinct sets of neurons are activated by the same stimuli following odor onset and offset. Such dissimilar responses were not even evoked by different odorants.

Further, we found that although both ON and OFF responses evoked strong PN responses, only the ON responses evoked oscillatory field potentials. Prior work (Stopfer et al., 1997), supported also by our data reveal that blocking fast GABAergic inhibition from local neurons to projection neurons completely eliminate odor-evoked oscillatory field potentials (new data; refer **Rebuttal Figure 4; new Supplementary Fig. 6b–d**). This pharmacological manipulation clearly shows GABAergic inhibition is needed for creating oscillations. Since these LFP oscillations are not present following stimulus termination, we conclude that recurrent inhibition is disengaged during the OFF response epochs. These results are strongly supported by our modeling work that replicates these *in vivo* observations (**Fig. 7**).

More importantly, our results clearly show that palp-closing responses occur only during epochs when odor-specific OFF responses were observed (also supported now by additional experiments and analyses).

We want to highlight this particular statement made by Reviewer4 that succinctly and beautifully captures our contribution:

“I think that the importance of the manuscript lies in making the community think about behavior termination as an active process that needs active neurons to turn it off. In that light, the manuscript makes an important contribution.” – Reviewer 4

Therefore, we strongly believe that this is a fundamental study to understand the ensemble neural correlates of behavioral dynamics. This, in fact, is the first study that focuses on explaining the termination of conditioned odor-evoked responses. Our results show that ‘unsensing’ is an active neural process and proposes a simple mechanism to generate odor-specific OFF responses. This study also answers how different phases of odor-evoked neural responses play a role in shaping the behavioral dynamics (onset, sustained, and reset).

Finally, the reviewer is indeed correct that the similarity between locust olfaction and primate audition are phenomenological. However, we think it is an important one. Our results highlight that sensory circuits of distinct modalities, complexities, evolutionary origin and position in information processing hierarchy may employ similar encoding principles to represent presence and absence of stimuli.

Specific Comments:

1. The "millisecond" in the beginning of the 6th row in Introduction should be plural.

Done

2. Figure 1a: Maybe should also quantify the responses as normalized to the pre-stimulus period? For example, the PN2 "peak" response to 1% 2oct on is not really an "on" response, which should actually be inhibition.

We divided the firing rates in different time bins by the peak firing rate. This limited the firing rates between [0, 1]. We employed this normalization to emphasize the differences between the ON and OFF responses and allow comparison between neurons. It is worth noting that this normalization is done only for generating the response color bar that allows qualitative comparisons in **Fig. 1** only. All other analyses (e.g. neural trajectory and classification analysis) take into account raw spike counts (in 50 ms time bin) during the entire ON and OFF periods.

To be consistent, we defined ON and OFF time windows for all 4 s long odor stimulus as following:

- ON response epoch is defined as the 4 s time window from odor stimulus start.
- OFF response epoch is defined as the 4 s time window from the termination of odor stimulus.

We are not trying to classify these responses. We just wanted to show peak firing responses during these two time windows, and reveal how non-overlapping neurons were activated during these two response periods.

3. Figure 1b: Left: What's the scale for the mean firing rate? Please include in the figure. Right: the "spokes" are distracting. Is the steady state fixed point same as the origin or just masked by the spokes? Maybe code time as the color intensities of the trajectories?

1. A color bar has been added to the plot. Note that the firing rates were log transformed to allow comparison across neurons.
2. The steady state fixed point is close to the baseline/origin but not the same. It is masked due to the orientation of the trajectory. We chose ‘spokes’ to highlight different planes of evolution of the ON and OFF trajectories and to be consistent with our previously published works (Saha et al., 2013; Saha et al., 2015). However, Reviewer1 has raised an important point. We have now added a supplementary figure showing the same trajectories without spokes (**Supplementary Fig. 4a**). As recommended, we have color-coded time evolution to highlight the differences between the steady state and the baseline.

Rebuttal Figure 1: Ensemble projection neuron responses are visualized after the dimensionality reduction using principal component analysis. Each axis corresponds to one of the top three principal components that best captures the variance in the dataset. Percentages of variance captured are shown along each axis. The color bar shown on top reveals how time since odor onset is represented in these trajectory plots. “B” indicates the baseline or pre-stimulus activity. In all panels, steady-state ON responses and OFF responses are identified using red and blue dotted circles, respectively. PN response trajectories during the ON and OFF durations are plotted for the following four odors: hex 1%, 2oct 1%, iaa 1% and bzald 1%.

4. Figure 1f: Does the 2s time window start at odor onset and offset or in the middle?
 Yes, the 2 s window for analysis shown in **Fig. 1f** starts at the odor onset and offset.

5. Figure 2: The observation that the most difference is seen between the on and off periods of the same odorant might just be due to the inhibition during the odor on period preceding the prominent off responses. Does the observation generalize to the odorants other than the 2 shown?

We had shown that this observation is true for 2 pairs of odorants: **Fig. 3e**, hex-2oct and **Fig. 3f**, bzald-iaa. We could only do the comparisons between a pair of odorants as hex-2oct and bzald-iaa were collected from different sets of PNs. However, we revisited our previously published data (Saha et al., 2013) and repeated analysis for three other odor pairs (see **Rebuttal Figure 2**). Note that the observed ON vs. OFF

responses of the same or different odorants are more distinct when compared to the ON responses of different odorants. This figure has been added to the supplementary section (**Supplementary Fig. 5d**).

Rebuttal Figure 2: Left, the response similarity (see Methods) between the apple 1% ON template with apple 1% ON, mint 1% ON, apple 1% OFF, and mint 1% OFF are shown (mean \pm s.d.). Asterisks indicate significant changes in similarity (* $P < 0.05$, paired t-test, $n = 10$ trials). Similar plots are shown on the right for cit 1% vs. ger 1% and 2hep 1% vs. chex 1%. Data was re-analyzed from a previous study (Saha et al., 2013). For comparisons between ON responses of the same odorant (i.e. first bar on each panel), a leave one trial out validation approach was used.

6. Figure 5: Why use 1s odor delivery instead of 4s as in the previous PN recordings? The absence of the observation of prolonged LN response to odor after delivery is not proof that no LN has prolonged response, given the small number of LNs recorded. This is at best just correlation from a limited sample size and does not justify the causality implied in the title and abstract. On the other hand, could the absence of LN odor-off response be explained by the absence of activity from ORNs rather than the disengagement from the PNs?

We chose a 1s duration for a couple of reasons. First, we note that all the necessary odor-evoked response dynamics that we focused in our work can be shown with a stimulus pulse of 1s duration. Our extracellular recording showed that both 0.75 s and 10 s duration hexanol pulses have a strong and consistent offset response (**Fig. 5**).

Second, to convince the reviewer that local neuron responses for a 1s pulse and a 4s pulse are similar, we have included the following figure (**Rebuttal Figure 3**, which is compiled from **Fig. 6b–d** and **Supplementary Fig. 6a** in the manuscript). Note that the LN responses last the duration of the odor pulse for both 1s and 4s long stimuli. Further, phase locking with simultaneously recorded LFP also happens only during the duration of the odor pulse in both cases.

Rebuttal Figure 3: (a) Top, intracellular response of a local neuron (LN2) and simultaneously recorded extracellular local field potential (LFP) are shown for a 1s long odor stimulation. Bottom left, power at different frequencies are shown for a LN before, during, and after an odor puff. Bottom right, Cross-correlations calculated between the local neuron membrane fluctuations (LN2) and the local field potentials are shown. (b) Same results are shown for a 4 s long odor pulse for the same local neuron. Notice that in both cases LN responses and LFP oscillations are limited to the odorant exposure period.

We note that published works have examined responses of individual projection and local neurons using 1s pulses rather than 4s pulses (Stopfer et al., 1997; Stopfer et al., 2003). So, it helps compare our results with those already published, while at the same time revealing the novelty of our observation.

We agree that we have provided only a handful of LN responses. However, prior work has already shown that engaging LN activity is important for generating LFP oscillations (Stopfer et al., 1997). Pharmacological blocking of fast GABAergic inhibition using picrotoxin injection reversibly abolishes these odor-evoked field potential oscillations during the odor ON period. We have repeated these experiments and included them as a supplementary result to further support our claim that LNs must be

disengaged following odor termination in order to evoke strong PN activities but without LFP oscillations (Supplementary Fig. 6b–d).

Rebuttal Figure 4: (a) Left, LFP oscillations are shown for a 4 s hexanol puff (control case; before picrotoxin bath application). Right, trial averaged frequency spectrogram is shown for the same odor stimulation ($n = 3$ trials; see Methods). (b) Same results are shown when 100 μ M picrotoxin was applied in saline bath to block the GABAergic local neurons' input to the PNs. (c) The LFP oscillations and the oscillatory power during odor presentation window recovers after saline wash. (d) Left, mean ORN firing rate plot is superimposed on the frequency spectrogram shown in panel a ($n = 24$ ORNs). Right, same plot as in left panel but with mean PN firing rate superimposed ($n = 80$ PNs).

Important observations:

1. Local field potential oscillations only last the duration of the odor pulse (i.e ON period). No oscillatory power after stimulus termination.
2. These oscillations can be abolished when fast recurrent inhibition from LNs are blocked using a bath application of picrotoxin (GABA_A receptor antagonist).
3. PN response after stimulus termination, although stronger than the persistent response, failed to evoke oscillatory field potential.
4. Sensory input from ORNs is weaker following stimulus termination.

Conclusion based on these experimental observations (also validated using the modeling study):

Recurrent inhibition from local neurons onto projection neurons is not present during OFF epoch (else LFP oscillation will also be evoked during this response window). Sensory input from ORNs onto LNs is important for engaging LN inhibition (note that PN activity is a poor indicator of whether or not LFP oscillations were present; **Rebuttal Fig. 4d**).

7. Figure 6: The modeling exercise only suggests a potential mechanism, which has not been proven to be actually true in this study.

We agree with the reviewer that we indeed used our modeling work to reveal a minimal model necessary to recreate our *in vivo* results and understand the relative contributions of the different components of the model. Some aspects of the model such as abolishing LFP oscillations by blocking fast GABAergic inhibition is already known and has been re-tested and included in the current version of the revised manuscript (see **Rebuttal Figure 4**).

A summary of the model components and their contributions is as follows:

1. LN to PN inhibition -> LFP oscillations
2. LN dependence on ORN input -> necessary to engage LNs only during the odor ON periods and limit LFP oscillations to these epochs. Note, ORN input is weak and almost comparable to baseline levels following odor offset (**Rebuttal Figure 4d**).
3. bi-directional adaptation of PN excitability -> necessary for creating strong PN OFF responses after odor termination when ORN input is weak.

We have done a bifurcation analyses to systematically study the importance of strengths of local neuron inhibition and bi-directional PN response adaptation for entraining oscillation field potentials and for generating distinct ON and OFF responses. These additional figures and analyses have been included in the main text of the revised manuscript (**Fig. 7**).

Figure 7

Rebuttal Figure 5: Local field potential activity (LFP; top trace) and six modeled projection neuron (PN) spiking activities are shown. Four different model architectures were evaluated:

(a) *Model architecture 1:* feed-forward ORN inputs to the local neurons (LNs) were removed. This made the total input received by LNs too weak and therefore the LNs were not activated when stimulus was introduced. As a result, PNs did not receive any feedback inhibition. Also, note that the stimulus-evoked oscillatory field potentials were not observed.

(b) *Model architecture 2:* LNs received inputs from both ORNs and PNs. As a result, LNs were activated and PNs received recurrent inhibition from LNs. Oscillatory field potentials were observed in this model during stimulus exposure period. However, the model did not generate a strong activity following stimulus termination (i.e. no 'OFF' responses).

(c) *Model architecture 3:* PN excitability was adapted in a bi-directional manner. LN inputs to PNs were removed. PNs did not receive feedback inhibition. Therefore, the model did not evoke stimulus-evoked LFP oscillations or strong PN responses following stimulus termination.

(d) *Model architecture 4:* PN responses were adapted in a bi-directional manner. LNs received inputs from ORNs and PNs. Therefore, LNs were activated by input stimulus and PNs received feedback inhibition. Therefore, the model produced stimulus-evoked oscillatory field potentials. The strong inhibition to a subset of PNs during the odor input increased the excitability of the inhibited PNs and thereby causing a strong OFF response in this model.

(e) Bifurcation analysis showing the relative importance of recurrent inhibition from LNs (y-axis) and bi-directional response adaptation (x-axis) for generating oscillatory local field potential in the 5 – 55 Hz frequency range. The horizontal banding reveals that the strength of the feedback inhibition alone is necessary and sufficient for generating LFP oscillations.

(f) Bifurcation analysis showing the relative importance of recurrent inhibition from LNs (y-axis) and bi-directional spiking threshold adaptation (x-axis) for generating distinct ON and OFF neural activities. Note that both strong recurrent inhibition and bi-directional spiking threshold adaptation are important for generating a distinct ON vs. OFF responses.

8. Figure 7: The authors used a CS- odor, benzaldehyde, partially overlapping with the CS+ odor, hexanol, to interfere near the end of the hex on response and the start of the hex off response. Here they appear to be over-interpreting the data. As shown in the bottom panel of Figure 7d, the population activity induced by the overlapping odors is quite complex, with the bzald-on peak pushing apart the hex-on and hex-off epochs, and the hex-off epoch overlapping with the second peak of the bzald-on epoch, finally with both hex-off and bzald-off epochs overlapping. The POR result for the overlapped odor sequence is not as clean cut as the authors stated either. The POR during hex-off/bzald-on seems clearly higher than both the baseline and the hex alone case, while the authors interpreted the first time point of the dip in palp separation as the start of palp closing without taking into account the sustained "half-opening" of the palps. Moreover, the seemingly same delay between the rise of hex-off probability curve and the first dip in palp separation does not warrant the strong statement of the stimulus-evoked off responses being REQUIRED for behavioral reset, which can only be addressed by somehow eliminating the odor off responses and seems impossible to do in the current system.

This is a terrific point! We thank the reviewer for noticing this and bringing it up. This result highlights and beautifully strengthens our conclusions as we discuss below.

The reviewer is indeed correct that the termination of OFF responses is slower for the hexanol overlap than solitary hexanol introductions (see **Rebuttal Figure 6**; top panels). If we examine the pattern match of the ensemble activities during these epochs with the hexanol ON and OFF templates (**Rebuttal Fig. 6**; middle panels), the following observations can be made. (i) for the overlapping presentation, the pattern match with hexanol ON template terminates as soon as benzaldehyde is introduced. (ii) In addition, neural activities following termination of hexanol when benzaldehyde was still present significantly reduced

pattern match with the hexanol OFF templates. Note also a biphasic OFF response which corresponds well to the two-step closing of the palps in behavior. Therefore, predictions of the POR by the ON-OFF model (**Rebuttal Fig. 6**; bottom panel) clearly reveals that the termination of the POR should be slower for the hexanol-benzaldehyde sequence than that hexanol solitary introductions.

This result clearly shows that the OFF response indeed is an excellent predictor of the palp closing dynamics and added as **Supplementary Fig. 10b,c**.

We once again thank the reviewer for raising this insightful point!

Rebuttal Figure 6: (a) Top, the median palp opening response is plotted \pm s.e.m; $n = 30$ Locusts) in the case of 4s duration hexanol presentation. Middle, the ensemble neural response match with the hex-ON and hex-OFF template are shown during the odor onset and offset periods. Bottom, the predicted POR using the ON-OFF model (see Methods) is shown for 4s hexanol solitary presentations. (b) Same format as in (a), but results are shown for a hex-0.5 s overlap-bzald odor sequence

9. Figure 8: There is no data on any local inhibitory interneurons so it is not clear whether the underlying mechanism is similar to that of the locust PNs.

The marmoset auditory ON-OFF responses was presented to reinforce what we saw in olfaction may be a general principle. Even in this auditory circuit, orthogonal ON-OFF responses were generated by the neural circuits without the need for distinct ON-OFF pathway (i.e. ‘ON’ tuned and ‘OFF’ tuned neurons as in vision). We agree with the reviewer that the similarities are at a phenomenological level but an

important one in our opinion. It highlights that some signal processing principles are conserved independent of circuit complexity (more vs. less # of neurons), position in the sensory information processing hierarchy, sensory modality (audition vs. olfaction) or evolutionary origin. It might be worth noting that our claims regarding mechanism are limited only to olfaction as no such data exists in marmoset A1 (however, a plausible hypothesis that remains to be tested is that similar mechanisms are at play in both circuits).

From a functional viewpoint, our observation highlights that even in auditory cortex, neural circuits generate distinct OFF responses, which in turn may contribute to active ‘unhearing’ of a sound (similar to ‘unsensing’ in olfaction)

10. Supplementary Figure 1: There are a lot more ORNs in the antennae than the PNs in the antennal lobe. It is fair to compare data from around 20 ORNs to that from 80 PNs. Maybe the absence of the odor off response is due to the small sample size for ORNs.

Although we have shown only a small amount of ORNs for this study, we examined our previous datasets from a larger collection of ORNs responses to an odor panel (**Rebuttal Fig. 7**). It is worth noting that population neural firing rates of these ORNs do not show a distinct OFF response to any odor in the panel. We believe that this data strongly supports our reasoning.

Rebuttal Figure 7: Mean ORN firing responses are shown for a panel of odorants. The shaded rectangle indicates the 4 s of odor delivery. The figure is modified from our previous study (Saha et al., 2013).

Reviewer #2 (Remarks to the Author):

This manuscript analyzes some interesting features of ON and OFF responses and considers the important question of what makes these patterns of neural activity differ, which is especially important in contexts in which behavior is driven at short latency and so can be completely explained by no responses. Unfortunately many of the data are presented in a non-quantitative way, despite considerable quantitative expertise of the PI and the conclusions are not well-supported by the data. Mechanisms to explain the differences in OFF and ON responses are insufficiently explored as well.

We thank the reviewer for finding our study important. We have now performed additional quantitative analyses and expanded our modeling work to address the concerns that this reviewer has raised.

The abstract and introduction of the paper make rather strong claims about the connection to behavior, which I find somewhat overstated. The authors identify a correlate of behavior, but there is nothing in the off responses that specifically have been identified as encoding features of the behavior, nor is the mechanism by which off responses differ in their ability to recruit inhibition etc well explained in the authors' results.

We showed that ensemble response pattern match with the OFF response template of the conditioned odorant is a good predictor of the behavioral POR reset. Our results indicated that perturbing the pattern match with ON response template by introducing a distractor odorant does not alter the behavioral POR responses after initiation (**Fig. 8d, e**). Whereas, as rightly pointed out by Reviewer 1, the termination of behavioral POR response does slow down when the distractor is introduced as the pattern match with the conditioned odor's OFF responses template reduces compared to the control case (i.e. solitary introductions). This is clearly highlighted in **Rebuttal Figure 6**.

Furthermore, we have now performed additional behavioral experiments with the pulsatile delivery of the conditioned odorants to amplify the connection between neural OFF responses and POR termination.

Regarding mechanistic insights, we have added models with different architectures to clearly illustrate:

- (i) how recurrent inhibition from local neurons are engaged by sensory input
- (ii) the role of recurrent inhibition to entrain field potential oscillations (pharmacological manipulations also added to validate and amplify our modeling results)
- (iii) the need for bidirectional adaptation of PN responses to generate strong OFF responses even though sensory input from ORNs is weak during this epoch.

We believe our newer results/analyses further lend support to our interpretation of the data and the mechanisms for generating specific type of ensemble responses during stimulus ON and OFF periods.

Finally, we note that we have added an additional analysis to show that neural response pattern matches (with both ON and OFF response templates) are Granger causal (at 5% significance level) with the behavioral POR responses.

The claim that in general ON responses for different odors are more similar than ON and OFF responses for the same odor should be further analyzed. Can "typical" features of OFF responses be identified? Perhaps an analysis of correlation would provide some insight?

This is an excellent suggestion. Our results showed that response similarity is much less between the ensemble ON response and OFF responses evoked by the same stimulus compared to the angular

similarities between the ON responses generated by different stimuli (**Fig. 3e,f** and **Supplementary Fig. 5d**).

Self and cross-correlational analyses:

Note that the angular similarities were measured based on the mean response vector during stimulus onset or offset. As recommended by the reviewer, we performed an analysis wherein we computed similarities/correlation between every ensemble vector (in 50 ms time bin) with those observed in ON and OFF periods of the same odor (self-correlation) or different odors (cross-correlation). These results are summarized in **Rebuttal Fig. 8** (also added as **Fig. 3** in the main manuscript)

As can be noted, correlations between ensemble response vectors during stimulus ON period are highly correlated amongst themselves (square blocks of high correlations). The same observations hold true for ensemble response vectors observed during the OFF epochs. Whereas correlations between response vectors observed during the ON periods with those evoked during OFF periods (i.e. off diagonal blocks) were extremely low. These observations were true when we compared the ON vs. OFF responses of the same odorants (top row), and when ON vs. OFF responses were compared across odorants (bottom row).

Rebuttal Figure 8: Correlation analyses between the ON and OFF responses.

(a) Schematic overview of the analysis approach. Each rectangular column indicates population neuron response vector in a 50 ms time bin. Right, self- and cross correlations between response vectors in different time bins were computed and shown as a color-coded image.

(b) Correlations between ensemble response vectors evoked by an odorant in different time bins following stimulus onset are shown. The 4s stimulus ON and 4s stimulus OFF periods are identified using red and blue bars along the axes. Spike counts were averaged across trials ($n = 25$ trials) and used for this analysis. Note that each pixel represents correlation between one ensemble vector with another. Similarly, one row or column represents the correlation between one ensemble vector with all other vectors in the identified time periods (80 ON response vectors and 80 OFF response vectors). The color scheme used for representing the correlation values is shown on the right; cooler colors indicate lower correlations; hotter color represents higher correlations. In general, the diagonal blocks tended to have higher correlations (more red pixels), whereas the off diagonal blocks had pixels mostly of lower correlations (i.e. more blue pixels).

(c) Similar correlation plots but comparing the ON and OFF response vectors of different concentrations of the same odorants are shown. Comparisons were made between 1% and 0.1% dilutions of the following four odorants: hexanol (hex), 2-octanol (2oct), isoamyl acetate (iaa), and benzaldehyde (bzald).

(d) Cross-correlations between different odorants are shown. Comparisons were made between the following four odor pairs: hex 1% and 2oct 1%, hex 0.1% and 2oct 1%, bzald 1% and iaa 1%, and bzald 0.1% and iaa 1%.

ON vs. OFF response motifs:

We examined if there are spiking patterns that are characteristic of the ON and the OFF responses. To do this we used a clustering algorithm and identified the top four response clusters. Note that the PN responses during the entire 8 s window following stimulus onset was used for this analysis (4s stimulus ON and 4s stimulus OFF; 160 dimensional response vectors). The top four response motifs identified were:

- A transient response following stimulus onset (transient ON)
- A persistent response lasting the stimulus duration (persistent ON)
- A transient response following stimulus offset (transient OFF)
- A persistent response with slower fall time constant following stimulus termination (persistent OFF)

These responses illustrate that the spiking patterns during the ON and the OFF periods are conserved but only the combination of neurons activated changes significantly (**Rebuttal Figure 9**).

Furthermore, we found that total number of spikes fired across all projections neurons recorded for different odorants during the ON and the OFF response windows were comparable with each other (**Rebuttal Figure 11**). In other words, similar number of spikes were generated using the ON and the OFF response periods.

However, the distribution of time-to-peak-response was sharper for the ON response window compared to the OFF responses (**Rebuttal Figure 10**). This result indicates that more ON responsive neurons reach their peak in a synchronized fashion compared to the OFF responsive neurons.

All of these results are now included in the **Supplementary Fig. 1**. of the revised manuscript.

Rebuttal Figure 9: Results from an unsupervised clustering analysis of olfactory projection neuron responses are summarized and shown here (see Methods for details). Two predominant response types (ON vs. OFF responses) were identified with two major sub-types for each case. Mean firing rates (\pm s.d.) averaged across projection neurons are shown as a function of time for each response cluster. Insets show all individual PN responses assigned to a particular cluster. n indicates number of PN responses assigned to a particular cluster.

Rebuttal Figure 10: The time-to-peak-response distributions for ON and OFF PNs are shown here (see Methods).

Supplementary Figure 4

Rebuttal Figure 11: (a) Total PN spike count (summed over all neurons recorded, $n = 80$ for hex and 2oct; $n = 81$ for bzald and iaa) during stimulus ON and stimulus OFF periods are shown for all four odorants used in the study (at 1% v/v). Comparison between ON and OFF spike counts are provided for the following integration window sizes: 1s, 2s, 3s, 4s (beginning from stimulus onset or stimulus offset). Parity between the ON and OFF responses is shown as a dotted line along the diagonal.

(b) Similar plot as in **panel a** but shown for the lower concentrations (0.1% v/v) of the same four odorants.

If so, what are they and how may they be explained mechanistically? Related to this is the observation that spike-field synchrony is reduced during the OFF responses even when avg firing rate is relatively high. The authors suggest that this reflects differential engagement of inhibition, but evidence in favor of this is not strong.

These are valid points that can be easily addressed. Most of the differences between the ON and OFF responses can be understood in the following way:

- ON responses – driven largely by strong sensory input from ORNs
- OFF responses – generated due to release from inhibition in the antennal lobe circuits

The expanded modeling work (see **Rebuttal Fig 5**) reveals the relative contribution of the different components of the model. To reiterate, recurrent inhibition when engaged generates oscillatory field potential. This can be reversibly abolished using GABA_A blocker (picrotoxin; see **Rebuttal Fig 4**). During the OFF response window, the PN firing rate is relatively high but there is no LFP oscillations. This is very similar to the pharmacological manipulations, and indeed blocking GABAergic inhibition makes the LFP field potentials during both ON and OFF periods relatively similar. These results clearly support our interpretation of differential engagement of inhibition during ON and OFF epochs. (also see response to Reviewer 1 on Pgs. 7 through 10 for additional details)

The authors' interpretation of these data to state that “neural encoding formats vary significantly” seems overly loaded and cumbersome. Without a consensus on what a neural encoding format means, this statement is not very meaningful.

By encoding formats being different we mean that spiking responses occurred with oscillatory synchronization or without it. Note that only during the ON response window the neural firing and LFP oscillations are observed, whereas during the OFF response window there is relatively high spiking activity but no oscillatory field potentials. We have clarified this in the manuscript.

For figure 2, some clearer, more intuitive way of displaying the data to emphasize the authors' point would be very helpful. I would recommend showing spike rasters but showing them in order of the change in firing rate for ON responses for one odor. This would facilitate the readers' understanding of the analysis. Moreover, some description of the approach to the analysis of the data and the degree of difference between ON and OFF responses vs different odor responses should be included.

This is a good suggestion. We found that other ways of showing the raw data to be equally confusing. So, instead, as alluded by this reviewer, we have replaced the old figure and included a correlation analysis (**Fig. 3**) to reveal the lack of similarity between ON and OFF responses during the entire 4s onset and offset time periods.

Generally more description of data means and significance should be included in the text. Statements like: “We found that in general, almost all PNs that were activated during stimulus exposure period were inhibited following stimulus termination with the firing activity reaching below baseline levels.” are extremely unsatisfying. Is this a significant effect or not? What does “almost all” mean? Some descriptive statistics would be helpful. What fraction of cells show increased firing rates during ON or OFF responses? What is the overall distribution of these responses? Simply showing mean changes in firing rate may disguise much of the interesting data.

In the revised manuscript, we have avoided such statements or supported them with more statistical details when necessary. We have added the following figure as new **Supplementary Fig. 1c**, where we clearly indicated the percentage distribution of ON and OFF transient and persistent PN responses for

individual odors tested. Distributions of PN responses were obtained from the cluster analysis used in **Rebuttal Figure 9**.

Rebuttal Figure 12: Percentages of neurons with a particular response motif: ON transient, ON persistent, OFF transient, OFF persistent are shown for all four odorants: hex 1%, 2oct 1%, iaa 1%, bzald 1%. Notice that this is a summary of the cluster analysis shown in the **Rebuttal Figure 9**.

How significant and/or surprising is this observation that the ON and OFF responses are approximately orthogonal? (pg 2) For example, if the authors took ON responses from one set of cells and appended them to OFF responses of a different set and then performed PCA, would they still observe this orthogonal relationship? Or not?

This is a terrific question! We thank the reviewer for raising this issue as it helps understand the neural data better. In general, we found that ON responses of different odorants to be more similar (i.e. less distinct). However, ON vs OFF responses of different odorants were also found to be more distinct. We have added this revised figure in the main manuscript (**Fig. 3e,f**).

Rebuttal Figure 13: A comparison of response similarity (see Methods) between ON and OFF response segments of the same and different odorants are shown. Left, similarity with respect to hex 1% ON template is shown (mean \pm s.d.). Asterisks indicate significant change in similarity ($*P < 0.05$, paired t-tests with Bonferroni correction for multiple comparisons, $n = 25$ trials). Right, similar plots are shown but now comparing the response similarity with respect to the bzald 1% ON template.

The trajectory analyses showing both ON and OFF responses of two different odorants also illustrate this observation clearly. Note that hex-ON and 2oct-ON are similar compared to ON vs. OFF responses of hex and 2oct. The same qualitative results hold true for a different odor pair, iaa-bzald as well.

Rebuttal Figure 14: (a) Population response trajectories of PNs ($n = 80$) are plotted after the dimensionality reduction using PCA. Both ON and OFF responses are shown for two odorants: hexanol and 2-octanol. Percentages of variance captured are shown along each axis. **(b)** ON and OFF response trajectories of isoamyl acetate and benzaldehyde are shown.

I expected the section labeled “Information Content Analysis” to include some information theoretic analysis of the population activity such as a quantification of total and conditional entropy. How many bits of information are available from the ON vs the OFF responses? Authors discuss anecdotally the information content of off responses as in “In sum, these results suggest that OFF responses are as consistent as the ON responses, and they actively convey information about the termination of a particular stimulus at a specific intensity.” Quantitative assessment of these claims is critical to include in the body of the MS.

We have re-labeled this section and apologize for misleading the reviewer.

However, we have also performed an information theoretic analysis as recommended by this reviewer. In brief, we estimated the information content carried by the spike trains of neurons during ON and OFF durations using the mutual information rate (Shannon, 1948) between odor stimulus and the neural response. We used the “direct method” approach (Reinagel and Reid, 2000; Strong et al., 1998) by finding the difference between the total and conditional entropies of the response.

$$I(S; R) = H_{total} - H_{noise}$$

Mutual information rate estimation was calculated for each neuron independently, and we found that the mean information rate ($n=80$ neurons) as 7.34 bits/s and median as 5.4 bits/s for the ON response. For the OFF response, we found the mean information rate as 5.13 bits/s and median as 2.85 bits/s. This suggests that the amount of information conveyed by the OFF responses is somewhat lower than those observed during the ON period. The distribution of information content for all the neurons during both ON and OFF responses are plotted in **Rebuttal Figure 15**. However, we found that the information rate

distributions of ON and OFF responses were statistically indistinguishable (two-sample Kolmogorov-Smirnov test, $k = 0.1625$, $P < 0.05$). We have also added this analysis as **Supplementary Fig. 3**.

In sum, this result supports our interpretation that only a few features distinguish the stimulus-evoked ON and OFF responses: (i) combinations of neurons activated, (ii) whether or not oscillatory field potentials were entrained, and (iii) the latency with which projection neurons reached their peak responses

Rebuttal Figure 15: Information theoretic analysis of ON and OFF responses.

(a) The estimated information rate for ON response is plotted for 80 neurons (hex-2oct odor pair). Responses of each neuron to five unique stimuli were used to estimate the total entropy (see Methods). The variations observed in the twenty five repeated trials of the same stimulus were used to estimate the noise entropy. (b) Estimated information rate for the OFF response is plotted for the same 80 neurons. (c, d) Histograms of ON and OFF information rate distributions are plotted, respectively.

The features of the model that are important for mediating the kinds of responses observed are not clearly tested, lessening the impact of the model. Some mechanistic predictions would make the model of greater interest and strengthen the manuscript as a whole.

We have now updated our modeling study to include contributions of different mechanisms and components towards OFF response generation (see **Rebuttal Figure 5**). There are two mechanistic predictions made by our modelling work.

First, our modeling work predicts that a strong sensory input is necessary for engaging recurrent inhibition through local neurons. Two pieces of evidence support this mechanistic insight. Our ORN recordings verify that sensory input from antenna was weak during these OFF response epochs even though PN responses were relatively strong.

Second, we repeated experiments to block GABAergic inhibition using pharmacological agent and showed that in the absence of this recurrent inhibition, the oscillatory power in the field potential oscillations during odor ON period becomes negligible during the both ON and OFF response epochs (see **Rebuttal Figure 4**).

These two pieces of evidences suggest that LN inhibition causes field potential oscillations but are engaged only when strong sensory input from ORNs is also present.

Next, our model predicts that to evoke a relatively high projection neuron spiking activity in the absence of strong ORN input, a bi-direction adaptation of PN spiking responses is necessary. This prediction remains to be tested experimentally.

Reviewer #3 (Remarks to the Author):

Saha and colleagues present an important new set of observations about the representation of odor onset and offset in the insect antennal lobe, taking advantage of the electrophysiological accessibility of the locust antennal lobe to carefully measure circuit-level properties. The core result is that the patterned activation of projection neurons by removal of neurons is a key component of ensemble encoding of behaviorally relevant sensory information.

All of the experiments are beautifully designed and presented. I am persuaded by all of the arguments, and I have only minor comments that might help make the paper more understandable to its readership.

We thank the reviewer for the positive feedback. We believe we have done all the suggested experiments proposed by this and other reviewers and addressed every single one of the concerns raised. We hope that the reviewer will find the much-improved manuscript suitable for publication.

My understanding is that the "OFF" responses of the PNs is intimately connected to and shaped by the inhibitory circuit. Post-inhibitory rebound is not a new phenomenon, and understanding precisely how it plays a role in the antennal lobe is a significant contribution. Towards this end, it would be good to spend more time describing the model and simulation. A diagram of the model with pertinent synaptic connections would be helpful, as it would clarify the fact that ORNs activate LNs, LNs inhibit ORNs and PNs, and PNs activate LNs.

The reviewer is indeed correct. Our modeling reveals that recurrent inhibition and post-inhibitory rebound (albeit a little more persistent form of rebound) is sufficient to replicate all our *in vivo* observations. As recommended, we have now revised and expanded our modeling work to clarify the contributions of the different components of the model better (see **Rebuttal Figure 5**). We hope that these additions will make our points clearer and highlight the contributions of the modeling work.

Illustrations of key dynamical features within the diagram would also help. For example, it would help to convey the "bidirectional adaptive threshold mechanism" that has been posited to occur in the projection neurons, which is apparently fundamental to bringing the ON responses in line with the OFF responses. As far as I can tell, that such a mechanism exists is a prediction of the model that remains to be tested. It is unclear to me that the "bidirectional adaptive threshold mechanism" is really what generates the orthogonal ensemble activities, although it would certainly promote it.

The reviewer is indeed correct. The 'bidirectional adaptive threshold mechanism' is a prediction from the model that remains to be tested. We performed bifurcation analyses where we systematically varied the relative importance of either recurrent inhibition or spike-threshold and found that a combination of these two is necessary for creating a strong OFF responses. In the absence of inhibition, the lowering of spike threshold following hyperpolarization that we incorporated in our model was not engaged and therefore failed to elicit strong OFF ensemble response. These clarifications have been added to the revised manuscript.

In the recent Nagel and Wilson (2016) paper in the Journal of Neuroscience, local neurons in *Drosophila* were shown to have both "ON" and "OFF" responses, and a similar PCA analysis showed that the first two principal components separately described these responses. If their LN dynamics were thus plotted in the same way as the locust PN data, I would also expect orthogonal representations.

We agree! This is what we would expect if the LNs also had strong ON and OFF responses. However, for locusts, we have not found evidence yet that indicate LNs can have strong OFF responses (although our sample size is too small, we use the generation of LFP oscillations as an indirect metric here). Single unit LN responses remain to be tested in a systematic manner. Nevertheless, this is an important reference which was published while our paper was in review and has been added to the manuscript. We thank the reviewer for bringing this manuscript to our attention.

The authors alternately refer to "angular similarity" and "angular distance" when referring to the same thing. Be consistent.

We apologize for the inadvertent confusion caused. We have fixed this issue in the revised manuscript.

The authors design a very clever odor associated learning experiment to test the behavioral relevance of the "OFF" response. They train the animal with a certain odor until the animal initiates feeding behavior when the odor is presented and terminates feeding when the odor is removed. The key experiment is what happens when adding an untrained odor to create a mixture before the trained odor is removed. Animals terminate the response when the trained odor is removed, and not when the "template match" to the trained odor is erased.

We are glad the reviewer liked the behavioral experiment using an overlapping sequence of two odorants. This is an excellent summary of the experiment and result.

The underlying assumption, I think, is that there is a single classifier at work. But what would happen if there are multiple classifiers, each sensitive to different subsets of PNs? I think this could be arranged to generate the same result, allowing the template match to the trained odor to be largely preserved in one classifier even when a new odor is added to create the mixture.

That is correct! We have assumed that the classifier at work uses ensemble activity for computing pattern matches and mismatches. The reason we think that multiple classifiers sensitive to different subsets of PNs may not exist stems from two pieces of evidence, which are not directly included in this manuscript.

1. As shown below, when responses of more than five different odorants are compared, our results indicate that the number of uniquely responding subset of PNs to a single odorant (hexanol 1%) becomes vanishingly small (~ 2–3 %). This would make any classifier based on subsets of uniquely responding neurons untenable.

Rebuttal Figure 16: Percentages of neurons uniquely responding to hexanol alone is shown. The percentage of unique responders reduces when comparisons across odors are made. For example, hex vs.

(2oct, iaa) indicate number of neurons that exclusively responsive to hex and not to 2oct or iaa. These results are part of a separate ongoing study and are only included in the rebuttal.

- In a recent study that was published in Nature Neuroscience (Saha et al., 2013), we found geraniol attracted locusts in a T-maze assay whereas citral repelled locusts in the same T-maze assay. A mixture of geraniol and citral, also attracted locusts in the T-maze. The degree of attraction could be predicted based on the overall pattern match. We note that even in this case, there was a subset of citral specific PNs that were responding to the mixture. However, the overall behavior was better predicted when ensemble activities were used. The figure and caption are reproduced from our previous study (Saha et al., 2013).

Rebuttal Figure 17: Behavioral results validate predictions from physiology data. (a) Results from a T-maze assay are shown. Each locust was given 4 min to make a decision: i.e. select a T-maze arm, reach and touch the sidewall at the end of the selected arm with its leg or antenna. Bar plots showing preferences of locust to the three test stimuli delivered: citral or geraniol or a mixture of geraniol and citral. Preference index is defined as the percentage of locusts choosing the odor arm minus the percentage choosing the mineral oil arm. Overall, locusts were repelled by citral but attracted by geraniol and the geraniol–citral odor sequence. The response to geraniol and the geraniol–citral sequence were both significantly different from the citral response (exact binomial test; * $P < 0.05$ with Bonferroni correction for multiple comparisons; $n = 20$ locusts for each case) **(b)** PN classification probabilities for the same set of stimuli used in the T-maze experiments are shown.

One might test if there is only one classifier at work, by delivering and removing the mixture all at once. If the animal fails to respond, then the animal would not be recognizing the presence of the trained odor within the mixture by a more specialized classifier. If it does respond, then it would be picking out the addition and removal of the trained odor irrespective of the distracting odor.

There is an important clarification we need to make regarding odor mixtures. Our earlier study showed that mixtures whose components are presented simultaneously (no time lag) are processed differently than those presented in a sequential fashion (Saha et al., 2013). In asynchronously presented mixtures, the novel component generally dominates the mixture responses. This is true in our hex-0.5s overlap-bzald mixture.

We performed the experiment suggested by this reviewer. We collected neural datasets examining the responses to the binary mixture of hexanol and benzaldehyde. We also performed behavioral experiments where we examined the responses of locust trained with hexanol and tested with both hexanol (conditioned stimulus) and a synchronous mixture of hexanol and benzaldehyde.

In contrast to the asynchronous mixture (previous data), the PN ensemble response to the mixture of hexanol and benzaldehyde where both components were delivered simultaneously were dominated by the hexanol responses and the ensemble activity pattern matched better with the hexanol ON response templates. Note that the pattern match with the ON responses of hexanol was indeed weaker for the binary mixture. We found that when conditioned locusts were presented with this mixture, the POR responses were also weaker than the response to the conditioned stimulus presented solitarily. These results again reveal how the degree of pattern match is a good indicator of the degree of the POR responses and is a nice supplement to our main results (**Supplementary Fig. 9**). We thank the reviewer for this excellent suggestion!

Rebuttal Figure 18:

(a) Neural response trajectory evoked by the overlapping sequence of hex-bzald is shown. The black trajectory shows the 4s period following bzald application. The response trajectories elicited by solitary presentations of hex and bzald are also shown to facilitate comparisons. Color-coded numbers on the plot indicate time since the introduction of a particular odorant. n represents the number of neurons used for this analysis.

(b) Results from a bin-by-bin, trial-by-trial classification analysis are shown for the hex-0.5s overlap-bzald overlapping sequence. The ON and OFF responses observed during solitary hex and bzald

introductions were used as templates to be pattern-matched. Bottom, the probabilities of pattern-match with different response templates are shown as a function of time. Boxed region identifies a small time segment starting just before the distractant (bzald) onset and ending after the termination of the conditioned stimulus (hex). Same figure as shown in **Fig. 8d**.

(c) Neural trajectory for the mixture of hex and bzald where both components were introduced simultaneously (note only ON responses are shown for clarity). The synchronous binary mixture of hex-bzald generated population neural responses that were aligned more with the hex-ON response alone. This new data was collected from a different PN population as compared to **panel a, b**.

(d) Similar plot as in **panel b** but now showing neural pattern match for synchronous mixture of hex and bzald.

(e) Median palp-opening responses are shown for a 6 s hexanol presentation (CS, in red) and the synchronous binary mixture (hex+ bzald, in black). Error bars represent s.e.m. Total of 27 locusts were tested for this study.

(f) No significant palp opening response was observed when a 6 s bzald odorant was puffed solitarily.

Reviewer #4 (Remarks to the Author):

In the manuscript “Engaging and disengaging recurrent inhibition mediates sensing and unsensing of a sensory stimulus” is an interesting study. In this study, authors make three major points (in my opinion) – 1) That odor ON and OFF activate different populations of neurons, 2) That, in the locust, this orthogonality might result from different degrees of recurrent inhibition engaged by odor-ON and odor-OFF. 3) That neurons that mediate ON response result in behavior initiation. More interestingly, the neurons that are activated during OFF are important for behavior termination. I think that the importance of the manuscript lies in making the community think about behavior termination as an active process that needs active neurons to turn it off. In that light, the manuscript makes an important contribution. The experiments are well-executed and the analysis is insightful.

We thank the reviewer for finding our work interesting and this concise summary of our contribution.

I do have 3 critiques. The first critique is aimed at further strengthening the evidence for the behavioral experiment. The other two critiques mostly relate to streamlining the manuscript.

1. RELATION BETWEEN NEURAL RESPONSE AND BEHAVIOR: The authors have done a reasonable job at convincing me of their result that the neurons that are turned OFF drive response termination. But, I am not totally convinced. The following things might help: i) I have several general questions about the behavior itself. I am quite surprised that up to 10 s, the locust keeps its palp open. Does it make sense that the locust keeps its palps open for 10s? I imagine that if one kept the odor on for even longer, it would eventually shut its palp. Have the authors explored this issue? Also, how does this result compare to other classical conditioning paradigms?

The reviewer is correct on multiple counts. Yes, it is surprising that the locusts keep their palps open for the entire odor duration. We do not have POR data when odor lasted for more than 10 s.

The asymptotic learning rates of locusts (~60-70% of trained locusts learn the task in 6 trials) compare favorably with other insects also trained using classical conditioning assays such as proboscis extension reflex (in bees, fruit flies, and moths). However, to our knowledge, no quantitative data to examine behavioral response dynamics exists.

Another general question is that there is clearly a really long tail to the behavior. That is, the palp separation is greater even 10 seconds after the odor is turned off (Figure 7C). I doubt that that long tail can be explained by the dynamics of the neurons in the antennal lobe. And that’s OK. We all know that there are several steps between the antennal lobe and behavior.

The POR return to baseline is slow as the reviewer correctly points out. We also agree with the reviewer that higher order processing centers may play a role in generation of that long POR tail. However, as we indicated in our response to Reviewer 1 (see **Rebuttal Figure 6**), changes in POR return to baseline correlate well with the OFF responses strength and dynamics. In particular, it is worth noting that the two major phases of POR reduction after termination of hexanol (in the hex-0.5s overlap – bzald sequence) occur following the time periods when a big increase in pattern match with hexanol OFF response were observed. Therefore, we propose that OFF responses of projection neurons are a good predictor of the behavioral response termination dynamics.

I would suggest finding a method to subtract the long tail. The exact method would depend on whether this is seen before conditioning or only after conditioning. This is important because I think it will have a great effect on the fits.

We did not subtract the tail as we wanted to present raw data as much as possible.

ii) It appears that the ON model is a better fit to the data in Figure 7C.

Both ON and ON-OFF model fit the solitary odor evoked POR well but we agree with the reviewer that the ON model does marginally better. However, the ON model performance degrades for the hex-0.5 s-bzald overlap condition. That latter was the main point we were making in **Fig. 8**. Comparison between different models also confirms this observation (**Fig. 8g**, **Supplementary Fig. 8c**).

iii) I like the experiment with benzaldehyde. It is quite creative. But, I am not sure I fully understand it. Why does the addition of benzaldehyde change the hex-off. I am also wondering why the authors did not do the much simpler experiment of giving several pulses of hexanol. Compare 4s pulse of hexanol to 2s-ON-2s-off-2s-ON.

We wanted to test how perturbing the ON/OFF responses of the conditioned odor (hex) by a distractant odor (bzald; does not induce POR on its own) changes the behavior. That is why we introduced benzaldehyde 0.5 s before hexanol termination. Our previous study showed that the introduction of a novel odor during the steady state of a background odor switch the ensemble neural activity towards the novel odor response (Saha et al., 2013). Indeed, when benzaldehyde was introduced after hexanol, the ensemble neural response lost pattern match with the hex-ON template and gained similarity with the bzald-ON template (**Rebuttal Fig. 18**). However, after hexanol termination, the population activity again switched to gain pattern match with the hex-OFF templates. In this way, we introduced a gap between hexanol ON and OFF template match, which we compared with the palp closing. The main reason was to introduce a delay between the hex ON and OFF response to decouple their functional role.

As recommended by this reviewer, we have now performed the 2s-ON-2s-off-2s-ON experiment in order to further test our hypothesis. When multiple pulses of hexanol were presented, we noted that PN ON-OFF responses faithfully followed stimulus dynamics. As can be expected, the POR responses followed the dynamics of the neural responses. The palps opened during time bins when pattern matches with hexanol ON template were observed and closed following pattern match with hexanol OFF responses. Therefore, these results further lend strong support to our interpretation of our data.

As we mentioned before, the need for the more complex stimulus sequence comes from the fact that both ON and ON-OFF model can predict the results observed to the pulsatile stimuli of the hexanol (CS) alone. Nevertheless, these newer behavioral experiments suggested by the reviewer led to important findings and have now been added in the manuscript (**Fig. 9**).

Figure 9

Rebuttal Figure 19: Neural and behavioral responses to pulsatile delivery of an odorant.

(a) Left, ensemble PN response trajectory is shown for 6 s long hexanol pulse. ON and OFF trajectories are identified using red and blue colors, respectively. Right, PN response trajectory is shown for 2s ON–2s OFF–2s ON hexanol pulse. Red and purple portions of the trajectories indicate the ensemble ON activity during the first and second hexanol pulse (ON responses), and the blue/cyan trajectories trace the PN ensemble responses following the termination of the first and the second hexanol pulse.

(b) Results from a classification analysis are shown in a bin-by-bin, trial-by-trial fashion. Based on the closest template, a class label has been assigned for each 50 ms time bin. hex ON response template – ‘red tick mark’, hex OFF response template – ‘blue tick mark’, and time bins when the ensemble neural activities that differed significantly from both hex ON and OFF response templates were labeled using a ‘black tick mark’. Classification results for ten trials are shown for 6s hex pulse and 2s ON–2s OFF–2s ON hexanol pulse. A leave-one-trial-out validation was used for generating these results. The colored bar on the top indicates stimulus exposure periods.

(c) The probabilities of pattern match with hex-ON and hex-OFF templates are plotted as a function of time for an uninterrupted 6s hexanol pulse and a 2s ON–2s OFF –2s ON hexanol pulses.

(d) Behavioral palp opening responses are plotted (median \pm s.e.m) for hex 6s and hex 2s ON–2s OFF–2s ON pulses. The prediction from the ON-OFF model (purple trace) is also shown for comparison.

2. INTRODUCTION AND GENERAL PITCH: I found the introduction to be too broad and all encompassing and found myself not agreeing with many of the statements. I suggest that the authors should start with the simple premise that in most systems there are neurons that turn ON when the stimulus is ON and others that turn ON when the stimulus is OFF and that they are going to explore that motif.

We have made minor revisions to the introduction but our pitch is still focused on the surprising distinctiveness between the ON and OFF responses and their relevance to shaping the behavioral response dynamics. If the reviewer still has issues, we will be happy to revise again if the reviewer can further pinpoint what he/she thinks are not agreeable to this reviewer.

3. MECHANISM FOR ORTHOGONALITY: I am a little confused as to what the authors think is critical for orthogonality. They mention that “bidirectional adaptive threshold” was important. I think that that is all one needs to generate orthogonality. Essentially PNs downstream to ORNs that are inhibited will be turned PN when the odor is turned off. This occurs in every sensory system, and this is all you need. In that sense, the evidence for recurrent inhibition is cool but specific to locust. The phenomenon, itself, is likely to be ubiquitous.

We have clarified the contributions of recurrent inhibition and bidirectional adaptive threshold in generating the data observed. In short, they are both needed to evoke a strong OFF response after odor termination when sensory input from ORNs is weak. This is clearly demonstrated in the revised and expanded modeling results (**Rebuttal Figure 5**). We agree with the reviewer that phenomenon of bidirectional adaptive threshold and resulting distinct ON vs OFF responses may exist in other sensory systems as well.

References:

- Reinagel, P., and Reid, R.C. (2000). Temporal coding of visual information in the thalamus. *J Neurosci* 20, 5392-5400.
- Saha, D., Leong, K., Li, C., Peterson, S., Siegel, G., and Raman, B. (2013). A spatiotemporal coding mechanism for background-invariant odor recognition. *Nat Neurosci* 16, 1830-1839.
- Saha, D., Li, C., Peterson, S., Padovano, W., Katta, N., and Raman, B. (2015). Behavioural correlates of combinatorial versus temporal features of odour codes. *Nat Commun* 6.
- Shannon, C.E. (1948). A Mathematical Theory of Communication. *At&T Tech J* 27, 379-423.
- Stopfer, M., Bhagavan, S., Smith, B.H., and Laurent, G. (1997). Impaired odour discrimination on desynchronization of odour-encoding neural assemblies. *Nature* 390, 70-74.
- Stopfer, M., Jayaraman, V., and Laurent, G. (2003). Intensity versus identity coding in an olfactory system. *Neuron* 39, 991-1004.
- Strong, S.P., Koberle, R., van Steveninck, R.R.D., and Bialek, W. (1998). Entropy and information in neural spike trains. *Phys Rev Lett* 80, 197-200.

Reviewers' Comments:

Reviewer #1 (Remarks to the Author):

The revised manuscript includes additional data and analysis that make it stronger. However, the gap between the wording throughout the manuscript and the actual experimental evidence remains and the manuscript still often confuses correlation with causation. For example, the title emphasizes the role of recurrent inhibition in “sensing” and “unsensing” of sensory stimuli, seemingly implying the absence of recurrent inhibition is directly related to the generation of OFF response and behavior, whereas what the data actually shows is that the recurrent inhibition is required for oscillatory activity during the odor ON period and oscillation is absent during odor OFF, without any direct biological evidence showing the causality between the termination of recurrent inhibition and the generation of OFF responses in the projection neurons and in palp closing.

Specific responses to the rebuttal:

1. The authors keep stating the absence of LFP oscillation proves the absence of recurrent inhibition from local neurons onto projection neurons. This is a misinterpretation of causality because the only evidence is that recurrent inhibition is needed for oscillation. For example, if LFP oscillation is generated by recurrent inhibition and another component, both necessary, then oscillation can still fail if there is recurrent inhibition but without the other necessary component.
2. The inclusion of the bi-directional adaptation of PN excitability and the bifurcation analyses are important additions to the modeling section, although the authors should still stress the modeling exercise just helps suggest a mechanism and maybe even new predictions but cannot be used to draw conclusions as if it is a fact.
3. The authors' interpretation of the biphasic pattern match between odor OFF response and palp closure in the hexanol overlap experiment does suggest a tighter correlation than first glance, and is actually stronger than the odor pulse experiment. However, there is still no direct evidence that OFF responses alone can drive palp closing or they are necessary.

Reviewer #2 (Remarks to the Author):

The authors have thoroughly revised their manuscript and I am satisfied by this new version.

Reviewer #3 (Remarks to the Author):

The authors respectfully and thoroughly attended to all of my concerns. The paper is much improved, and makes a solid contribution to the field.

Reviewer #4 (Remarks to the Author):

The authors have done a fantastic job with the revision

Response to reviewers' comments (round 2):

We thank all four Reviewers for their insightful and thorough comments, which we believe, improved the manuscript significantly. We would also like to take this opportunity to thank the editor for his suggested text revisions which further clarified our contributions.

Reviewer #1 (Remarks to the Author):

The revised manuscript includes additional data and analysis that make it stronger. However, the gap between the wording throughout the manuscript and the actual experimental evidence remains and the manuscript still often confuses correlation with causation. For example, the title emphasizes the role of recurrent inhibition in “sensing” and “unsensing” of sensory stimuli, seemingly implying the absence of recurrent inhibition is directly related to the generation of OFF response and behavior, whereas what the data actually shows is that the recurrent inhibition is required for oscillatory activity during the odor ON period and oscillation is absent during odor OFF, without any direct biological evidence showing the causality between the termination of recurrent inhibition and the generation of OFF responses in the projection neurons and in palp closing.

We thank the reviewer for his comments. We have revised the manuscript to eliminate statements that imply causation. We have also revised title of the paper:

“Engaging and disengaging recurrent inhibition coincides with sensing and unsensing of a sensory stimulus”

We have also included changes suggested by the editor throughout the manuscript to suggest a mechanism rather than imply causation.

Specific responses to the rebuttal:

1. The authors keep stating the absence of LFP oscillation proves the absence of recurrent inhibition from local neurons onto projection neurons. This is a misinterpretation of causality because the only evidence is that recurrent inhibition is needed for oscillation. For example, if LFP oscillation is generated by recurrent inhibition and another component, both necessary, then oscillation can still fail if there is recurrent inhibition but without the other necessary component.

We are fully aware of the point the reviewer is raising. The failure of a bicycle to propel when the front wheel is removed does not make the removed component solely responsible for the function lost! However, the reviewer has failed to consider existing literature that we extensively cited that have already concluded the following:

“In conclusion, local neurons with extensive arbors monosynaptically inhibit projection neurons in the antennal lobe by the way of fast, PCT-sensitive, GABA-containing synapses. This inhibition underlies the synchronization of ensembles of projection neurons and thus the odor-evoked LFP oscillations in the mushroom body.”

-excerpt reproduced as is from MacLeod and Laurent (1996), Distinct mechanism for synchronization and temporal patterning of odor-encoding neural assemblies, *Science*, vo. 274, pp. 976-979.

These results have been further reiterated in the (Stopfer et al., 1997):

“We have previously shown that picrotoxin (PCT) applied to the locust antennal lobe selectively blocks the fast inhibitory synapse between local and projection neurons and abolishes their oscillatory synchronization.”

Therefore, the result that recurrent inhibition from local neurons to projection neurons is needed to entrain LFP oscillations has been well-established by previous studies (MacLeod and Laurent, 1996; Stopfer et al., 1997). We were not trying to prove this result in our work. Rather, we have only extended that result. During the off response periods, projection neuron spiking responses were elicited without local neurons being active and in the absence of local field potential oscillations. Therefore, we respectfully disagree with the point raised by the reviewer.

2. The inclusion of the bi-directional adaptation of PN excitability and the bifurcation analyses are important additions to the modeling section, although the authors should still stress the modeling exercise just helps suggest a mechanism and maybe even new predictions but cannot be used to draw conclusions as if it is a fact.

We have clearly indicated that the modelling work suggests a mechanism (Pg. 6, second paragraph):

“Hence, our modeling study suggests that stimulus-dependent engagement and disengagement of recurrent inhibition in the antennal lobe circuits provides a simple mechanism for generating distinct ON and OFF neural activities with differing response formats (oscillatory vs. non-oscillatory).”

3. The authors’ interpretation of the biphasic pattern match between odor OFF response and palp closure in the hexanol overlap experiment does suggest a tighter correlation than first glance, and is actually stronger than the odor pulse experiment. However, there is still no direct evidence that OFF responses alone can drive palp closing or they are necessary.

We agree that our results provide a tight correlation between neural responses and the behavior produced. This is why we included a Granger causality analysis to show that POR responses can be better predicted with neural pattern matches observed during the ON and OFF epochs than those made using POR response histories alone.

We argue that in distributed systems where several components contribute to behavior (complex systems even as simple one as a bicycle), causality will be difficult to convincingly prove! However, the point raised is well taken, and our text has been appropriately revised as recommended by the editor.

Page 2: “Further, our results reveal that switching between distinct neural ensembles over time is temporally correlated with the behavioral dynamics evoked by a stimulus. Notably, our results suggest

that such representations provide a potential mechanism for sensory neural networks to meet the evolving demands on the behavioral output during these epochs”

Page 7: “Taken together, these results strongly support our hypothesis that orthogonal neural activities may underlie opposing behavioral responses in this olfactory system.”

Reviewer #2 (Remarks to the Author):

The authors have thoroughly revised their manuscript and I am satisfied by this new version.

Reviewer #3 (Remarks to the Author):

The authors respectfully and thoroughly attended to all of my concerns. The paper is much improved, and makes a solid contribution to the field.

Reviewer #4 (Remarks to the Author):

The authors have done a fantastic job with the revision

We thank the Reviewers 2, 3 and 4 for helping us substantially enhance our manuscript.

References:

MacLeod, K., and Laurent, G. (1996). Distinct mechanisms for synchronization and temporal patterning of odor-encoding neural assemblies. *Science* 274, 976-979.
Stopfer, M., Bhagavan, S., Smith, B.H., and Laurent, G. (1997). Impaired odour discrimination on desynchronization of odour-encoding neural assemblies. *Nature* 390, 70-74.